# FERRET: FEDERATED FULL-PARAMETER TUNING AT SCALE FOR LARGE LANGUAGE MODELS

## ABSTRACT

Large Language Models (LLMs) have become indispensable in numerous real-world applications. Unfortunately, fine-tuning these models at scale, especially in federated settings where data privacy and communication efficiency are critical, presents significant challenges. Existing methods often resort to parameter-efficient fine-tuning (PEFT) to mitigate communication overhead, but this typically comes at the cost of model accuracy. To address these limitations, we propose *federated full-parameter tuning at scale for LLMs* (Ferret), the first first-order method with shared randomness to enable scalable full-parameter tuning of LLMs across decentralized data sources while maintaining competitive model accuracy. Ferret accomplishes this through three aspects: *(1)* it employs widely applied first-order methods for efficient local updates; *(2)* it projects these updates into a low-dimensional space to considerably reduce communication overhead; and *(3)* it reconstructs local updates from this low-dimensional space with shared randomness to facilitate effective full-parameter global aggregation, ensuring fast convergence and competitive final performance. Our rigorous theoretical analyses and insights along with extensive experiments, show that Ferret significantly enhances the scalability of existing federated full-parameter tuning approaches by achieving high computational efficiency, reduced communication overhead, and fast convergence, all while maintaining competitive model accuracy.

## 1 INTRODUCTION

Recently, Large Language Models (LLMs) have become indispensable tools across a wide range of real-world applications, from natural language processing tasks like translation (Xu et al., 2024) and summarization (Van Veen et al., 2024) to more complex tasks such as code generation (Liu et al., 2024) and decision-making systems (Shao et al., 2023). The immense scale and versatility of LLMs make them highly valuable in practice, but they also introduce significant challenges, particularly when they are fine-tuned in federated settings. Federated Learning (FL) offers a decentralized approach to fine-tuning LLMs while retaining data on local clients to ensure privacy. However, while this approach effectively addresses privacy concerns, it also results in **prohibitive communication overhead** when the model parameters of LLMs scale to billions.

One of the straightforward strategies to mitigate the prohibitive communication costs in the federated tuning of LLMs is parameter-efficient fine-tuning (PEFT). PEFT methods (Hu et al., 2022; Wei et al., 2024) focus on fine-tuning only a subset of model parameters, which is able to significantly reduce the communication overhead between clients and a central server (Che et al., 2023; Zhang et al., 2023; Kuang et al., 2024; Zhang et al., 2024b). Despite the effectiveness in reducing bandwidth usage, this type of approach often compromises **model accuracy** (Pu et al., 2023), as fine-tuning a subset of model parameters may fail to fully capture the nuances of local data distributions. Thus, recent efforts, e.g., FedKSeed (Qin et al., 2024), have been devoted to utilizing zeroth-order optimization (ZOO) (Nesterov & Spokoiny, 2017; Berahas et al., 2022) in federated full-parameter tuning of LLMs, aiming to maintain competitive model accuracy while reducing the communication overhead by transmitting only thousands of scalar gradients per round between clients and a central server. Unfortunately, this approach often suffers from its **poor scalability**, including **increased computational cost** per round and **a larger number of communication rounds** required for convergence, compared to FL methods that use first-order optimization (FOO), e.g., FedAvg (McMahan et al., 2017).

To this end, we propose *federated full-parameter tuning at scale for LLMs* (Ferret), the first first-order FL method with shared randomness to enable scalable federated full-parameter tuning of LLMs with *compelling computational efficiency, reduced communication overhead*, and *fast convergence speed*, all while maintaining *competitive model accuracy*. Ferret achieves this through three aspects: First, it employs widely applied first-order methods to perform computationally efficient local updates on each client, which typically requires fewer iterations to achieve the same local update process compared to existing ZOO-based FL. Next, Ferret projects these updates into a low-dimensional space, leading to a significantly reduced communication cost compared to existing FOO-based FL. Finally, Ferret reconstructs local updates from the low-dimensional space with shared randomness for effective full-parameter global aggregation, ensuring fast convergence and competitive model accuracy compared to existing ZOO-based FL. We further complement Ferret with rigorous theoretical analyses and principled insights, showing the theoretical advantages of Ferret over other baselines and guiding the best practices for its implementation. Finally, through extensive experiments, we verify that Ferret significantly outperforms existing methods with superior scalability and competitive model accuracy, making it a desirable solution for deploying LLMs in large-scale federated environments.

To summarize, our contributions in this work include:

- We propose Ferret, the first first-order FL approach with shared randomness (to the best of our knowledge), which significantly enhances the scalability of federated full-parameter tuning of LLMs while maintaining competitive model accuracy.
- We present rigorous theoretical analyses and insights to support the effectiveness of our Ferret, demonstrating its theoretical advantages over other baselines and guiding its best practices.
- Through extensive experiments, we demonstrate that Ferret consistently improves over existing methods in practice, offering both superior scalability and competitive model accuracy.

## 2  PROBLEM SETUP

In this paper, we consider the federated full-parameter tuning of an LLM using decentralized data $\{\mathcal{D}_i\}_{i=1}^N$ on $N$ local clients while preserving data privacy, i.e., without sharing raw data. Specifically, given a loss function $\ell(\cdot;\cdot)$, we aim to minimize a global objective $\mathcal{L}(\mathbf{w})$ defined as the average loss across $\{\mathcal{D}_i\}_{i=1}^N$ over the model parameters $\mathbf{w} \in \mathbb{R}^d$ of an LLM. That is,

$$\min_{\mathbf{w}} \mathcal{L}(\mathbf{w}) \triangleq \frac{1}{N} \sum_{i \in [N]} \mathcal{L}^{(i)}(\mathbf{w}) \quad \text{where} \quad \mathcal{L}^{(i)}(\mathbf{w}) \triangleq \mathbb{E}_{\mathbf{x}^{(i)} \in \mathcal{D}_i} \left[ \ell(\mathbf{w}; \mathbf{x}^{(i)}) \right] . \tag{1}$$

Following the practice in federated learning (FL), (1) can be solved through multiple rounds of local training and global aggregation. In each communication round, each client $i$ independently updates its local model parameters by minimizing its local objective $\mathcal{L}^{(i)}(\mathbf{w})$ based on its local data $\mathcal{D}_i$. After local training, the clients transmit their updated local model parameters to a central server, where they are aggregated to form an updated global model. This updated global model is then redistributed to all clients, and the process is repeated over rounds.

The main challenge in LLM federated full-parameter tuning is to ensure the **computational efficiency and the convergence speed** of the global model while **reducing the communication overheads**, particularly given that the parameter size $d$ of LLMs often reaches billions. While existing first-order FL (McMahan et al., 2017; Li et al., 2020; Karimireddy et al., 2020) can ensure compelling computational efficiency and convergence speed by applying first-order updates, they typically incur $\mathcal{O}(d)$ communication overheads due to the need to transmit the entire set of model parameters between clients and the central server. This type of methods hence is impractical for LLM federated full-parameter tuning due to the enormous size of LLMs. In contrast, although zeroth-order FL (Qin et al., 2024) can reduce these communication costs by transmitting only several scalar gradients from their finite difference-based gradient estimation with shared randomness, they often incur more computational cost to achieve the same local update progress and a larger number of communication rounds to converge compared with first-order FL. These naturally raise the question:

> *Can we combine the strengths of these methods to achieve scalable federated full-parameter tuning of LLMs with high computational efficiency, reduced communication overhead, and fast convergence?*

---

**Algorithm 1:** Federated Full-Parameter Tuning at Scale for LLMs (Ferret)

---

**Input:** Pre-trained model parameters $\mathbf{w}_0$, $N$ clients, number of rounds $R$, number of local updates $T$, number of bases $K$, local learning rate $\eta$

1 **for** each round $r \in [R]$ **do**
2     **for** each client $j \in [N]$ in parallel **do**
3        **if** $r > 1$ **then** // Step ①: Global Aggregation
4           Receive seeds $\{s^{(i)}\}_{i=1}^{N}$ and coordinates $\{\gamma_k^{(i)}\}_{i=1,k=1}^{N,K}$ from the central server
5           Generate random bases $\{\mathbf{v}_k^{(i)}\}_{i=1,k=1}^{N,K}$
6           $\mathbf{w}_{r-1} \leftarrow \mathbf{w}_{r-2} - \sum_{i \in [N]} \left( \sum_{k=1}^{K} \gamma_k^{(i)} \mathbf{v}_k^{(i)} \right) / N$
7        $\mathbf{w}_{r,0} \leftarrow \mathbf{w}_r$
8        **for** each iteration $t \in [T]$ **do** // Step ②: Local Updates
9           $\mathbf{w}_{r,t}^{(j)} \leftarrow \mathbf{w}_{r,t-1}^{(j)} - \eta \nabla \ell(\mathbf{w}_{r,t-1}^{(j)}; \mathbf{x}_{r,t-1}^{(j)})$ on randomly sampled data $\mathbf{x}_{r,t-1}^{(j)}$
         // Step ③: Projected Updates
10       Randomly choose seed $s^{(j)}$ and generate random bases $\{\mathbf{v}_k^{(j)}\}_{k=1}^{K}$
11       $\Delta_r^{(j)} \leftarrow \mathbf{w}_{r-1}^{(j)} - \mathbf{w}_r^{(j)}$, compute coordinates $\{\gamma_k^{(j)}\}_{k=1}^{K}$ based on (6)
12       Send $s^{(j)}$ and $\{\gamma_k^{(j)}\}_{k=1}^{K}$ to the central server

---

# 3 THE FERRET ALGORITHM

To answer this question, we introduce Ferret, *federated full-parameter tuning at scale for LLMs*, in Algo. 1. We present an overview of our Ferret algorithm in Sec.3.1, followed by a detailed explanation of the key techniques in Ferret in Sec. 3.2.

## 3.1 OVERVIEW OF FERRET

To achieve scalable LLM federated full-parameter tuning, our Ferret algorithm combines the strengths of both first-order FL, which offers efficient computation and fast convergence, and zeroth-order FL, which reduces communication overhead. Specifically, Ferret *(a)* follows first-order FL to apply first-order optimization methods for local updates on clients, ensuring both computational efficiency and fast convergence, and *(b)* draws inspiration from zeroth-order FL by projecting updates into a low-dimensional space using random bases that can be regenerated using shared randomness among clients for the reconstruction of these updates, thereby reducing communication overhead.

Our Ferret algorithm operates by repeating the following three sequential steps over many communication rounds, denoted by $r \in [R]$, where $R$ is the total number of rounds. For simplicity, we omit the subscript $r$ from the seeds, random bases, and projected coordinates in our notation.

**Step ①: Global Aggregation (Line 3-6 in Algo. 1).** At the beginning of the first round ($r = 1$), each client initializes its local model parameters using the pre-trained model parameters $\mathbf{w}_0$, i.e., $\mathbf{w}_1 \leftarrow \mathbf{w}_0$. For subsequent rounds ($r > 1$), each client $j \in [N]$ receives the random seeds $s^{(i)}$ and the corresponding $K$ projected coordinates $\{\gamma_k^{(i)}\}_{k=1}^{K}$ of every client $i \in [N]$ from the previous round. These random seeds (i.e., shared randomness) are then used to generate $d$-dimensional random bases $\{\mathbf{v}_k^{(i)}\}_{k=1}^{K}$ for each client $i$. [1] These random bases, along with the corresponding projected coordinates $\{\gamma_k^{(i)}\}_{k=1}^{K}$, are applied to reconstruct local updates as $\widetilde{\Delta}_{r-1}^{(i)}$ in every client $i$. The global model is then updated by aggregating these local contributions as follows:

$$\mathbf{w}_{r-1} \leftarrow \mathbf{w}_{r-2} - \frac{1}{N} \sum_{i \in [N]} \widetilde{\Delta}_{r-1}^{(i)} \quad \text{where} \quad \widetilde{\Delta}_{r-1}^{(i)} \triangleq \sum_{k \in [K]} \gamma_k^{(i)} \mathbf{v}_k^{(i)} . \tag{2}$$

**Step ②: Local Updates (Line 7-9 in Algo. 1).** After **Step ①**, each client $j$ will perform $T$-iteration first-order optimization on its local loss function by using the randomly sampled data for its local updates. Formally, if stochastic gradient descent with a local learning rate $\eta$ is used, the update rule

---

[1] Similar to (Qin et al., 2024), we can obtain $K$ random seeds from a single seed $s^{(i)}$ and employ these $K$ seeds to generate $K$ random bases independently for each client $i$. So, one seed is sufficient for each client.

for client $j \in [N]$ at iteration $t \in [T]$ of round $r \in [R]$ can then be represented as:

$$\mathbf{w}_{r,t}^{(j)} \leftarrow \mathbf{w}_{r,t-1}^{(j)} - \eta \nabla \ell \left( \mathbf{w}_{r,t-1}^{(j)}; \mathbf{x}_{t-1}^{(j)} \right) \quad \text{where} \quad \mathbf{w}_{r,0} \leftarrow \mathbf{w}_r . \tag{3}$$

Different from the zeroth-order update in (Qin et al., 2024) that requires many local update iterations, the first-order update in (3) enables each client to efficiently and effectively adapt the global model $\mathbf{w}_r$ to its specific data using a small $T$, thereby enhancing both the computational efficiency of this local update. Here, (3) can be implemented using any gradient method variant, e.g., Adam (Kingma & Ba, 2014).

**Step ③: Projected Updates (Line 10-12 in Algo. 1).** After completing the local updates above, each client $j$ randomly chooses a single new seed $s^{(j)}$ to generate $K$ new random bases $\{\mathbf{v}_k^{(j)}\}_{k=1}^K$ and employ these $K$ new random bases to project the local update $\Delta_r^{(j)}$ into a $K$-dimensional coordinates $\{\gamma_k^{(j)}\}_{k=1}^K$ based on the techniques in Sec. 3.2. Seed $s^{(j)}$ and projected coordinates $\{\gamma_k^{(j)}\}_{k=1}^K$ are then shared with other clients to facilitate the next round of global aggregation. By sharing only a single random seed and $K$ projected coordinates among $N$ clients where random bases $\{\mathbf{v}_k^{(j)}\}_{k=1}^K$ can be regenerated for global aggregation as shown in **Step ①** above, the communication overhead in LLM full-parameter tuning is therefore considerably reduced compared with first-order methods (e.g., FedAvg) especially when $T \ll d$. Of note, the communication of this seed $s^{(j)}$ can be mitigated if the same seed is used across all rounds $r \in [R]$, which can further reduce the communication overhead.

## 3.2 Update Projection and Reconstruction

As mentioned before, we aim to project the local updates into $K$-dimensional coordinates ($K \ll d$) to substantially reduce the communication overhead in LLM full-parameter tuning. To accomplish this, let $\Delta \in \mathbb{R}^d$ denote any local update, and let $\mathbf{V} = [\mathbf{v}_1 \ \mathbf{v}_2 \ \cdots \ \mathbf{v}_K] \in \mathbb{R}^{d \times K}$ represent the $K$ random bases generated by any random seed $s$, we solve the following convex minimization problem to determine the $K$-dimensional projected coordinates $\boldsymbol{\gamma} = [\gamma_1 \ \gamma_2 \ \cdots \ \gamma_K]^\top$:

$$\boldsymbol{\gamma} \triangleq \arg\min_{\mathbf{y}} \|\mathbf{V}\mathbf{y} - \Delta\| . \tag{4}$$

As $\mathbf{V}$ is singular with $K \ll d$, the close-form of $\boldsymbol{\gamma}$ and its corresponding reconstruction $\widetilde{\Delta}$ will be

$$\boldsymbol{\gamma} = (\mathbf{V}^\top \mathbf{V})^{-1} \mathbf{V}^\top \Delta, \quad \widetilde{\Delta} = \mathbf{V}(\mathbf{V}^\top \mathbf{V})^{-1} \mathbf{V}^\top \Delta . \tag{5}$$

**Choice of Random Bases V.** Particularly, if $\mathbf{V}$ is a rectangular matrix with ones on its main diagonal, meaning that each $\mathbf{v}_k$ is a standard basis vector, (5) simplifies to $\boldsymbol{\gamma} = \mathbf{V}^\top \Delta$, which then corresponds to a block-wise dimension selection for local update projection and reconstruction. However, this approach significantly reduces the number of parameters updated per round as $K \ll d$, potentially hindering the overall tuning performance. We thus propose to sample each element in $\mathbf{v}_k$ ($k \in [K]$) independently from a normal distribution with bounded 2-norm, i.e., $\|\mathbf{v}_k\| \leq 1$, aiming to realize and stabilize full-parameter tuning of LLMs for competitive overall performance. To achieve this, we can sample from a truncated normal distribution: $\mathrm{v} \sim \mathcal{N}(0, 1)$ with $\mathrm{v} \in [-1/\sqrt{d}, 1/\sqrt{d}]$ instead. The efficacy of this bounded norm will be demonstrated in Sec. 4.1 shortly.

**Reconstruction w/o Inversion.** Unfortunately, (5) incurs a computational complexity of $\mathcal{O}(K^2 d + K^3)$ and storage complexity of $\mathcal{O}(Kd)$ owing to the inversion of $\mathbf{V}^\top \mathbf{V}$ in (5), which is prohibitively costly, especially when $K$ is large and $d$ reaches billions. Since $\mathbf{V}^\top \mathbf{V}$ is a scaled empirical covariance for the aforementioned distribution of an identity covariance matrix (Vershynin, 2012), we propose to approximate $\mathbf{V}^\top \mathbf{V}$ with $\mathbf{I}_K$ (i.e., $K \times K$-dimensional identity matrix) and (5) as

$$\boldsymbol{\gamma} \approx (\rho K)^{-1} \mathbf{V}^\top \Delta . \tag{6}$$

Here, $\rho \triangleq 1 - \frac{2\psi(1/\sqrt{d})/\sqrt{d}}{2\Phi(1/\sqrt{d}) - 1}$, where $\psi(\frac{1}{\sqrt{d}})$ and $\Phi(\frac{1}{\sqrt{d}})$ is the probability density function (PDF) and cumulative distribution function (CDF) of the standard normal distribution evaluated at $1/\sqrt{d}$, respectively. This approximation leads to improved computational complexity of $\mathcal{O}(Kd)$ and storage complexity of $\mathcal{O}(\max\{K, d\})$, where the storage complexity is reduced due to the in-place operations

on random bases $\{\mathbf{v}_k\}_{k=1}^K$ when computing $\{\gamma_k\}_{k=1}^K$ sequentially. Consequently, we can reconstruct the true update $\Delta$ approximately using $\widetilde{\Delta}$ below

$$\widetilde{\Delta} = (\rho K)^{-1} \mathbf{V} \mathbf{V}^\top \Delta \,, \tag{7}$$

whose efficacy will be theoretically justified in Sec. 4.1. Finally, our (6) and (7) simplify the update projection and reconstruction in (5) into straightforward matrix multiplications.

**Block-Wise Reconstruction.** The computational complexity of $\mathcal{O}(Kd)$ and storage complexity of $\mathcal{O}(\max\{K, d\})$ for our reconstruction in (7) is still prohibitively costly, particularly for LLMs with billions of parameters. To address this, we propose a block-wise reconstruction technique to reduce both computational and storage complexities. Specifically, suppose the full dimension $d$ is divided into $L$ blocks, each with dimension $d_l$ such that $\sum_{l \in [L]} d_l = d$. Let $\Delta_l$ be the update for block $l$ and $K_l$ (with $\sum_{l \in [L]} K_l = K$) be the number of random bases allocated to this block. We propose to compute $\boldsymbol{\gamma}_l$ and reconstruct $\Delta_l$ using random bases $\mathbf{V}_l$ of dimension $d_l \times K_l$ as follows:

$$\boldsymbol{\gamma}_l = (\rho_l K)^{-1} \mathbf{V}_l^\top \Delta_l, \quad \widetilde{\Delta}_l = (\rho_l K_l)^{-1} \mathbf{V}_l \mathbf{V}_l^\top \Delta_l \,. \tag{8}$$

Here, $\rho_l \triangleq 1 - \frac{2\psi(1/\sqrt{d_l})/\sqrt{d_l}}{2\Phi(1/\sqrt{d_l})-1}$. This trick reduces the storage complexity to $\mathcal{O}(\max\{\{K_l, d_l\}_{l=1}^L\})$ that is straightforward to verify, and lowers the computational complexity to $\mathcal{O}(\sum_{l \in [L]} K_l d_l)$. Of note, (8) also significantly reduces the computational complexity of global aggregation compared to existing methods (Qin et al., 2024) (verified in Sec. 5). This block-wise reconstruction thus further enhances the scalability of our Ferret in the federated full-parameter tuning of LLMs.

# 4 THEORETICAL ANALYSES AND INSIGHTS

We now provide theoretical analyses to substantiate the effectiveness of Ferret: *(a)* reconstruction analysis in Sec. 4.1; *(b)* convergence analysis in Sec. 4.2; and *(c)* scalability and beyond in Sec. 4.3.

## 4.1 RECONSTRUCTION ANALYSIS

**Theorem 1** (Unbiased Reconstruction). *Given the reconstruction in* (7)*, we have*

$$\mathbb{E}\left[\widetilde{\Delta}\right] = \Delta \,.$$

To begin with, we demonstrate in Thm. 1 that our reconstruction in (7) is unbiased, with the proof provided in Appx. B.1. Of note, Thm. 1 shows that *(a)* the scalar $1/(\rho K)$ is crucial for (7) to achieve an unbiased reconstruction of the ground-truth update $\Delta$, and *(b)* our (7) avoids the bias commonly found in zeroth-order FL methods (Berahas et al., 2022), including FedZO (Fang et al., 2022) and FedKSeed (Qin et al., 2024). As a result, (7) is expected to provide a more accurate update reconstruction, which we will elaborate more below.

**Theorem 2** (Reconstruction Error). *Given the reconstruction in* (7)*, we have*

$$\mathbb{E}\left[\left\|\widetilde{\Delta} - \Delta\right\|\right] \leq \max\left\{2\sqrt{\frac{2\ln(2d)}{\rho K}}, \frac{2\ln(2d)}{\rho K}\right\} \|\Delta\| \,.$$

We then demonstrate the efficacy of our reconstruction in (7) by theoretically bounding the difference between the reconstructed update $\widetilde{\Delta}$ and the ground truth $\Delta$ in Thm. 2. The proof is in Appx. B.2. Of note, $1/\rho$ typically has an asymptotic rate of $\mathcal{O}(d)$, which we will verify empirically in Appx. C.4. Thm. 2 offers three critical insights of our Ferret: *(a)* Our reconstruction in (7) incurs a reconstruction error at a rate of $\widetilde{\mathcal{O}}(d/K)$ for $T$ local update iterations when $\sqrt{d} \geq K$, which generally aligns with the results in (Vershynin, 2010). This indicates that the reconstruction error of our (7) can be linearly reduced by increasing $K$. *(b)* Ferret avoids additional constant error items (Berahas et al., 2022) that are caused by the biased estimation in these zeroth-order FL methods, implying that our (7) can be more accurate. We will justify this further in our Thm. 3 below. *(c)* Thanks to the independence from the iterations (i.e., $T$) of local updates in Thm. 2, Ferret prevents the error accumulation over the local update iterations $T$, which is a common issue in zeroth-order FL methods (Fang et al., 2022; Qin et al., 2024).

**Theorem 3** (Connection with Zeroth-Order Method). *Define $g_k \triangleq \frac{\ell(\mathbf{w} + \epsilon \mathbf{v}_k; \mathbf{x}^{(i)}) - \ell(\mathbf{w}; \mathbf{x}^{(i)})}{\epsilon}$ where each element $\mathrm{v}$ in $\mathbf{v}_k$ is sampled from $\mathrm{v} \sim \mathcal{N}(0, 1)$ with $\mathrm{v} \in [-1/\sqrt{d}, 1/\sqrt{d}]$, $\boldsymbol{g} \triangleq [g_1 \cdots g_K]^\top$, and $\mathbf{V} \triangleq [\mathbf{v}_1 \ \mathbf{v}_2 \ \cdots \ \mathbf{v}_K] \in \mathbb{R}^{d \times K}$, assume $\ell(\cdot; \cdot)$ is $\beta$-smooth w.r.t its first argument, the zeroth-order reconstruction $\mathbf{V}\boldsymbol{g}/K$ used in (Fang et al., 2022; Qin et al., 2024) then incurs:*

$$\left\| \frac{1}{K} \mathbf{V}\boldsymbol{g} - \frac{1}{K} \mathbf{V}\mathbf{V}^\top \nabla \ell(\mathbf{w}; \mathbf{x}^{(i)}) \right\| \leq \frac{1}{2} \beta \epsilon .$$

We then show in Thm. 3 the connection between our update projection (6) and zeroth-order method used in (Fang et al., 2022; Qin et al., 2024). The proof is provided in Appx. B.3. Thm. 3 delivers three essential insights: *(a)* When $\epsilon \to 0$, the reconstruction $\mathbf{V}\boldsymbol{g}/K$ in zeroth-order method is equivalent to $\mathbf{V}\mathbf{V}^\top \nabla \ell(\mathbf{w}; \mathbf{x}^{(i)})/K$ and shares a similar form of (7) when $\Delta$ is replaced by $\nabla \ell(\mathbf{w}; \mathbf{x}^{(i)})$, implying that zeroth-order method in fact aims to approximate our reconstruction (7). *(b)* In practice, $\epsilon > 0$. So, zeroth-order method leads to a biased reconstruction with an additional error term of $\beta\epsilon/2$ compared to our (7), and this error will accumulate over $T$ local iterations, implying that our (7) can indeed be more accurate as we have demonstrated above. *(c)* In addition, zeroth-order method is typically coupled with a single gradient (i.e., $\nabla \ell(\mathbf{w}; \mathbf{x}^{(i)})$), whereas our (7) can be applied to any vector, making it more general. Overall, these results further verify the advantages of our (7) over the zeroth-order method used in (Fang et al., 2022; Qin et al., 2024), which we will also support empirically in Appx. C.4.

**Proposition 1** (Block-Wise Reconstruction Speedup). *For block-wise reconstruction (8) of size $L$,*

$$\sum_{l \in [L]} d_l K_l < \left( \sum_{l \in [L]} d_l \right) \left( \sum_{l \in [L]} K_l \right) = dK .$$

We next highlight the computational advantage of our block-wise reconstruction (8) in Prop. 1. The proof is in Appx. B.4. Prop. 1 indicates that by dividing the reconstruction of $d$-dimensional updates into smaller blocks $\{d_l\}_{l=1}^L$, we get a reduction in overall computational complexity that is strictly less than that of the full dimension $d$ in (7). E.g., when $d_1 = \cdots = d_L$ and $K_1 = \cdots = K_L$, we have $\sum_{l \in [L]} K_l d_l = Kd/L$, showing that our block-wise reconstruction (8) reduces the computational complexity of (7) by a factor of $1/L$. This implies that increasing the number of blocks $L$ can further enhance the computational efficiency of our block-wise reconstruction (8).

**Proposition 2** (Block-Wise Reconstruction Error). *For block-wise reconstruction (8) of size $L$, when $\sqrt{d_l} \geq K_l$ for any $l \in [L]$,*

$$\mathbb{E}\left[ \left\| \widetilde{\Delta} - \Delta \right\| \right] < \widetilde{\mathcal{O}}\left( \sum_{l \in [L]} \frac{\|\Delta_l\|}{\rho_l K_l} \right) ,$$

*which is minimized by choosing $K_l \propto \sqrt{\|\Delta_l\| / \rho_l}$.*

We conclude by analyzing the error induced by our block-wise reconstruction (8) and the corresponding optimal random bases allocation in Prop. 2. The proof is provided in Appx. B.5. Prop. 2 demonstrates that reconstruction error can be minimized by adaptively allocating the number of random bases according to the gradient norm of each block. This is intuitively reasonable because a larger gradient norm typically indicates a need for more immediate model updates in practice. Hence, this insight not only provides a theoretical foundation for optimizing Ferret but also offers practical guidance. That is, by aligning the number of random bases with gradient norms, practitioners can enhance reconstruction accuracy and overall model performance. This adaptive approach ensures efficient use of computational resources, making Ferret versatile and effective across different datasets and federated learning scenarios.

### 4.2 CONVERGENCE ANALYSIS

In this subsection, we present the convergence of Ferret in our Thm. 4 below when using stochastic gradient descent (SGD) for the local updates in (3). To simplify the analysis, we primarily focus on deriving theoretical results for a homogeneous setting, where $\mathcal{L}^{(i)}(\mathbf{w}) = \mathcal{L}(\mathbf{w})$ in (1). Results in the heterogeneous setting can be derived by following the same proof idea.

Table 1: Comparison of scalability (computation and communication per round, and #rounds to converge) and other factors (adaptability, generalization, and privacy). Here, $d \gg K \gg T$. Symbols: $\circ$ (fewer is better), $\heartsuit$ and $\diamond$ (more is better).

| Method | Type | Scalability | | | Others | | |
| --- | --- | --- | --- | --- | --- | --- | --- |
| | | Comp. | Comm. | #Rounds | Adapt. | Gen. | Privacy |
| FedZO | ZOO | $\mathcal{O}(\tau_0 K)$ | $\mathcal{O}(d)$ | $\circ \circ \circ$ | $\checkmark$ | $\heartsuit\,\heartsuit\,\heartsuit$ | $\diamond\,\diamond$ |
| FedKSeed | ZOO | $\mathcal{O}(\tau_0 K)$ | $\mathcal{O}(K)$ | $\circ \circ \circ$ | $\times$ | $\heartsuit\,\heartsuit\,\heartsuit$ | $\diamond\,\diamond\,\diamond$ |
| FedAvg | FOO | $\mathcal{O}(\tau_1 T)$ | $\mathcal{O}(d)$ | $\circ$ | $\checkmark$ | $\heartsuit\,\heartsuit\,\heartsuit$ | $\diamond$ |
| Ferret (ours) | FOO | $\mathcal{O}(\tau_1 T)$ | $\mathcal{O}(K)$ | $\circ \circ$ | $\checkmark$ | $\heartsuit\,\heartsuit\,\heartsuit$ | $\diamond\,\diamond\,\diamond$ |

**Theorem 4** (Convergence). *Define $D \triangleq \mathcal{L}(\mathbf{w}_0) - \min_{\mathbf{w}} \mathcal{L}(\mathbf{w})$. Assume that $\mathcal{L}(\mathbf{w})$ is $\beta$-smooth and non-convex, and $\mathbb{E}[\|\nabla\mathcal{L}^{(i)}(\mathbf{w}) - \ell(\mathbf{w};\mathbf{x})\|^2] \leq \sigma^2$ for any $\mathbf{x}, \mathbf{w}$, when choosing $\eta \leq \frac{1}{20\beta T}$ in Algo. 1, the following holds for federated full-parameter tuning with $\mathcal{L}^{(i)}(\mathbf{w}) = \mathcal{L}(\mathbf{w})$,*

$$\min_{r \in [R)} \mathbb{E}\left[\|\nabla\mathcal{L}(\mathbf{w}_r)\|^2\right] \leq \mathcal{O}\left(\frac{D}{\eta T R} + \eta T \sigma^2\right)$$

*where $[R)$ is the half-open interval $[0, R)$. Especially, by choosing $\eta = \frac{1}{20\beta T \sqrt{R}}$ in Algo. 1, the number of communication rounds are required to be $R = \mathcal{O}(1/\epsilon^2)$ to achieve an $\epsilon$ convergence error.*

Its proof is in Appx. B.6. Particularly, when $T = 1$, Thm. 4 recovers the result of standard SGD (Ghadimi & Lan, 2013). Thm. 4 provides three essential insights: *(a)* Thanks to our improved update reconstruction (7) as justified above, Ferret avoids the additional constant terms accumulated over $T$ local iterations, which are typically caused by the biased gradient estimation in zeroth-order FL methods (e.g., FedZO and FedKSeed) (Fang et al., 2022), thereby highlighting the superior advantage of Ferret over these zeroth-order FL methods in convergence speed. *(b)* Given a proper $\eta$, Ferret shares the same communication round complexity as SGD, at a rate of $\mathcal{O}(1/\epsilon^2)$, showing that the communication round complexity of Ferret is asymptotically comparable to that of standard SGD. *(c)* This communication rounds complexity is improved over that of zeroth-order FL methods (Fang et al., 2022) due to its independence from $d$ and other constant factors required by these zeroth-order FL methods, further highlighting the advantage of Ferret in communication round complexity and its improved efficacy in federated full-parameter tuning over these methods.

### 4.3 SCALABILITY AND BEYOND

With the theoretical results above, we summarize the scalability of Ferret and compare it to existing methods like zeroth-order FL (e.g., FedZO and FedKSeed) and first-order FL (e.g., FedAvg) in Tab. 1.

**Computation Per Round.** Of note, Ferret enjoys a computational complexity of $\mathcal{O}(\tau_1 T)$ for any client $i \in [N]$ per round, where $\tau_1$ is the per-iteration complexity of the first-order update (including forward and backward passes) in (3), and $T$ is the number of local iterations. This is comparable to the well-established FedAvg. In contrast, both FedZO and FedKSeed incur a complexity of $\mathcal{O}(\tau_0 K)$, with $\tau_0$ being the per-iteration complexity of the zeroth-order update (i.e., forward pass) and $K$ representing the number of forward passes. As first-order updates use more accurate gradients, $T$ will be smaller than $K$ (i.e., $T \ll K$) to attain the same local update progress. Although $\tau_1$ can be at most twice $\tau_0$, our Ferret is still more computationally efficient than FedZO and FedKSeed (see Sec. 5).

**Communication Per Round.** As only one seed and $K$ projected coordinates $\{\gamma_k^{(i)}\}_{k=1}^K$ from a client $i \in [N]$ need to be transmitted per round in Algo. 1 with $K \ll d$, Ferret incurs a communication overhead of $\mathcal{O}(K)$, which is similar to that of FedKSeed. This is significantly more efficient than FedAvg and FedZO, which have a communication complexity of $\mathcal{O}(d)$ due to their need to transmit the entire model (or gradients). This significantly reduced communication cost therefore makes Ferret especially suitable for federated full-parameter tuning of LLMs with billions of parameters.

**Rounds to Converge.** As revealed in Sec.4.2, our Ferret benefits from unbiased update reconstruction in (7) (validated in Thm. 1), enabling fast convergence with a small number of communication rounds to achieve $\epsilon$ convergence error (see Thm. 4). This is significantly more efficient than zeroth-order FL

Table 2: Comparison of Rouge-L (%) among various algorithms. Each cell reports the mean $\pm$ std of Rouge-L scores from the final round of four runs, each using a different random seed. All results, except for those pertaining to FedAvg and Ferret, are taken from (Qin et al., 2024).

| Algorithm | Natural Instructions | | Dolly-15K | |
|---|---|---|---|---|
| | DataJuicer-1.3B | LLaMA-3B | DataJuicer-1.3B | LLaMA-3B |
| FedPTuning | $19.61 \pm 2.71$ | $25.41 \pm 1.14$ | $23.98 \pm 3.23$ | $30.30 \pm 1.16$ |
| FedPrompt | $6.04 \pm 0.12$ | $8.95 \pm 2.47$ | $32.73 \pm 0.87$ | $24.50 \pm 4.78$ |
| FedIT-SGD | $19.40 \pm 1.83$ | $28.14 \pm 0.85$ | $27.23 \pm 0.68$ | $29.28 \pm 0.50$ |
| FedIT | $22.30 \pm 0.42$ | $28.13 \pm 0.50$ | $30.80 \pm 0.98$ | $33.23 \pm 1.51$ |
| FedZO | $21.74 \pm 1.91$ | $29.46 \pm 0.38$ | $32.91 \pm 0.67$ | $36.34 \pm 0.39$ |
| FedKSeed | $22.33 \pm 1.72$ | $29.77 \pm 0.75$ | $32.90 \pm 0.37$ | $35.64 \pm 0.83$ |
| FedAvg | $23.95 \pm 2.76$ | $32.11 \pm 0.70$ | $29.67 \pm 1.26$ | $30.98 \pm 1.66$ |
| Ferret (ours) | $24.99 \pm 0.99$ | $30.03 \pm 0.99$ | $30.63 \pm 0.84$ | $34.57 \pm 0.57$ |

methods like FedZO and FedKSeed, which require many more communication rounds to converge due to poor gradient estimation (Fang et al., 2022). FedAvg, applying the ground truth local update for its global aggregation, surely converges with the fewest rounds. Overall, Ferret remains a strong choice for federated full-parameter tuning of LLMs, even in terms of rounds to converge.

**Beyond Scalability.** Our Ferret also offers benefits in adaptability, generalization, and privacy. Unlike FedKSeed, which is limited to SGD, Ferret is highly adaptable, because both global aggregation (2) and local update (3) in Ferret can be implemented with any gradient method variant, e.g., the widely used AdamW (Loshchilov & Hutter, 2019) in LLM training. This adaptability thus makes it much easier to integrate Ferret into existing centralized tuning workflows for LLMs, facilitating a seamless transition to federated tuning. Besides, since Ferret enables federated tuning with full parameters, it is expected to deliver strong generalization performance as other federated full-parameter tuning methods like FedAvg, as supported in Sec. 5. Finally, by transmitting only seeds and low-dimensional projected coordinates among clients, rather than the entire model (or gradients) as in FedZO and FedAvg, Ferret ensures improved privacy for federated full-parameter tuning of LLMs.

Overall, Ferret strikes an optimal balance between computational efficiency, communication overhead, convergence speed, and other critical factors such as adaptability, generalization, and privacy. This makes it a highly scalable and desirable solution for federated full-parameter tuning of LLMs.

## 5 EXPERIMENTS

In this section, we evaluate the efficacy of our Ferret algorithm, following the practice in FedKSeed (Qin et al., 2024). We primarily compare Ferret with other federated full-parameter tuning baselines, including both zeroth-order methods (e.g., FedZO (Fang et al., 2022) and FedKSeed (Qin et al., 2024)) and first-order methods (e.g., FedAvg (McMahan et al., 2017)). Our evaluations use DataJuicer-1.3B (Chen et al., 2023) and LLaMA-3B (Touvron et al., 2023a) on the Natural Instructions (Wang et al., 2022) and Dolly-15K (Conover et al., 2023) datasets, as well as larger models (i.e., LLaMA2-7B and LLaMA2-13B (Touvron et al., 2023b)) on the CodeAlpaca (Chaudhary, 2023) and GSM8K (Cobbe et al., 2021) datasets. As demonstrated in Sec. 4.2, Ferret is guaranteed to converge faster than zeroth-order FL methods. Therefore, we run Ferret for fewer communication rounds compared to FedKSeed: 12 rounds versus 40 on Natural Instructions, and 20 rounds versus 60 on Dolly-15K. However, for more complex tasks such as CodeAlpaca and GSM8K, we run all algorithms, including Ferret, for 20 rounds to ensure a fair comparison. More experimental details and ablation studies are provided in Appx. C.1 and Appx. C.5, respectively.

### 5.1 COMPARISON ON ACCURACY

We present the model accuracy achieved by different federated tuning methods in Tables 2 and 3. The results in Table 2 demonstrate that federated full-parameter tuning methods (including FedAvg, FedZO, FedKSeed, and our method) generally achieve better model accuracy compared to PEFT-based federated tuning methods (such as FedPTuning, FedPrompt, FedIT-SGD, and FedIT). This

Table 3: More comparison of Rouge-L (%) among various algorithms. Each cell reports the mean $\pm$ std of Rouge-L scores from the final round of four runs, each using a different random seed.

| Algorithm | CodeAlpaca | | GSM8K | |
|---|---|---|---|---|
| | LLaMA2-7B | LLaMA2-13B | LLaMA2-7B | LLaMA2-13B |
| FedZO | $4.58 \pm 0.26$ | $6.19 \pm 0.32$ | $30.41 \pm 0.31$ | $13.63 \pm 0.34$ |
| FedKSeed | $8.33 \pm 0.98$ | $10.70 \pm 0.47$ | $28.26 \pm 3.60$ | $33.67 \pm 1.15$ |
| FedAvg | $15.41 \pm 0.43$ | $14.68 \pm 0.26$ | $38.30 \pm 0.40$ | $39.82 \pm 0.17$ |
| Ferret (ours) | $12.10 \pm 0.47$ | $11.84 \pm 0.91$ | $36.10 \pm 1.18$ | $34.50 \pm 1.42$ |

Table 4: Comparison of computational cost and communication overhead on LLaMA-3B, focusing on (a) the computational costs from local updates, global aggregation, and the overall tuning process; and (b) the per-round and overall communication costs. The improvement achieved by our Ferret is reported in brackets using blue (compared with FedKSeed) and orange (compared with FedAvg).

| Algorithm | Computational Cost (Sec.) | | | Communication Cost (# param.) | |
|---|---|---|---|---|---|
| | Local Update | Global Aggr. | Overall | Per-Round | Overall |
| FedZO | 32.6 | 0.3 | $1.3 \times 10^3$ | $6.0 \times 10^9$ | $2.4 \times 10^{11}$ |
| FedKSeed | 56.9 | 123.8 | $7.2 \times 10^4$ | $8.2 \times 10^3$ | $3.3 \times 10^5$ |
| FedAvg | 1.8 | 0.3 | 25.2 | $6.0 \times 10^9$ | $7.2 \times 10^{10}$ |
| Ferret (ours) | **5.6** $(10.2\times)$ | **24.7** $(5.0\times)$ | $\mathbf{3.6 \times 10^2}$ $(20.0\times)$ | $\mathbf{7.8 \times 10^3}$ $(10^6\times)$ | $\mathbf{9.4 \times 10^4}$ $(10^6\times)$ |

underscores the importance of full-parameter tuning for Large Language Models (LLMs). Importantly, the results in both tables show that our proposed method consistently delivers strong or competitive performance across four different scenarios. Specifically, on the Natural Instructions dataset, our method outperforms all others for different model sizes, with up to a 2.66% improvement over the next best method, FedKSeed. On the Dolly-15K dataset, our method maintains competitive performance. Moreover, on both the CodeAlpaca and GSM8K datasets, our method achieved noticeably improved accuracy over other zeroth-order baselines (i.e., FedZO and FedKseed). However, Ferret slightly underperform FedAvg, likely due to reconstruction errors caused by our method for these complex tasks. Overall, these results have well demonstrated the ability of our method to sustain strong model accuracy in practice across various datasets and model sizes.

## 5.2 COMPARISON ON SCALABILITY

Since we focus on federated full-parameter tuning of LLMs, we primarily provide a detailed scalability comparison of this type of methods, including FedZO, FedKSeed, FedAvg, and Ferret. We evaluate their scalability performance on Natural Instructions using LLaMA-3B (see Tab. 4) and GSM8K using LLaMA2-7B (see Tab. 5), where the calculation of computational cost and communication overhead is provided in Appx. C.2 and more comparison on LLaMA2-13B is in Appx. C.3. The results in Tab. 4 and Tab. 5 demonstrate that compared with FedKSeed, Ferret achieves substantial reductions in computational costs: a $10.2\times$ improvement for local updates on LLaMA-3B and $13.1\times$ on LLaMA2-7B, a $5.0\times$ improvement in global aggregation on LLaMA-3B and $5.8\times$ on LLaMA2-7B, as well as a $20.0\times$ improvement for overall tuning cost on LLaMA-3B and $6.8\times$ on LLaMA2-7B.[2] These advancements stem from several key innovations: our first-order local updates, which reduce the number of required iterations; block-wise reconstruction, which optimizes global aggregation; and precise reconstruction, which significantly decreases communication round complexity. Furthermore, compared to FedAvg that does not leverage any shared randomness, Ferret exhibits an enormous reduction in overall communication costs, i.e., $10^6\times$ on LLaMA-3B and $10^7\times$ on LLaMA2-7B. This emphasizes the ability of Ferret in scaling federated full-parameter tuning.

---

[2] The reduced improvement in overall tuning cost for our Ferret on LLaMA2-7B, compared to LLaMA-3B, is because that both Ferret and FedKSeed are using the same number of communication rounds for more complex tasks such as GSM8K.

Table 5: Comparison of computational cost and communication overhead on LLaMA2-7B, focusing on (a) the computational costs from local updates, global aggregation, and the overall tuning process; and (b) the per-round and overall communication costs. The improvement achieved by our Ferret is reported in brackets using blue (compared with FedKSeed) and orange (compared with FedAvg).

| Algorithm | Computational Cost (Sec.) | | | Communication Cost (# param.) | |
|---|---|---|---|---|---|
| | Local Update | Global Aggr. | Overall | Per-Round | Overall |
| FedZO | 54.1 | 0.7 | $1.1\times10^3$ | $1.4\times10^{10}$ | $2.8\times10^{11}$ |
| FedKSeed | 117.0 | 510.0 | $1.3\times10^4$ | $8.2\times10^3$ | $1.6\times10^5$ |
| FedAvg | 5.8 | 0.7 | $1.3\times10^2$ | $1.4\times10^{10}$ | $2.8\times10^{11}$ |
| Ferret (ours) | **8.9** (13.1×) | **88.3** (5.8×) | $\mathbf{1.9\times10^2}$ (6.8×) | $\mathbf{6.4\times10^3}$ ($10^6\times$) | $\mathbf{1.3\times10^4}$ ($10^7\times$) |

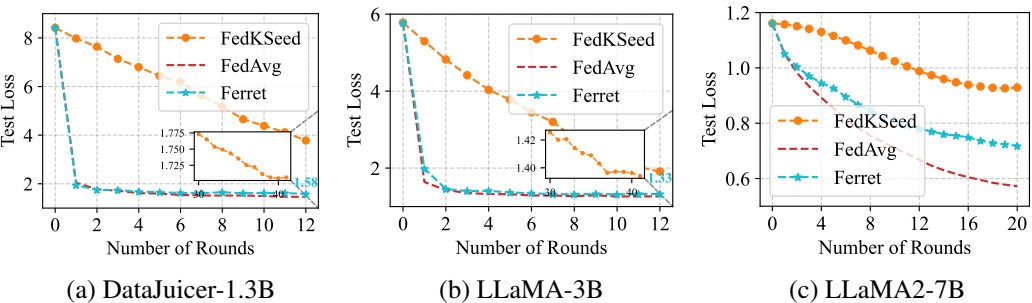

(a) DataJuicer-1.3B  (b) LLaMA-3B  (c) LLaMA2-7B

Figure 1: Comparison of communication rounds required by Ferret, FedKSeed, and FedAvg for convergence on Natural Instructions with (a) DataJuicer-1.3B and (b) LLaMA-3B, and (c) comparison on GSM8K with LLaMA2-7B.

In Fig. 1, we also compare the convergence speeds of Ferret with other baselines (e.g., FedKSeed and FedAvg) on Natural Instructions (with DataJuicer-1.3B and LLaMA-3B) and on GSM8K (with LLaMA2-7B). The findings show that, Ferret converges remarkably fast, requiring only two communication rounds in line with FedAvg compared to the 12 rounds needed by FedKSeed. This results in a 20× reduction in communication round complexity for both DataJuicer-1.3B and LLaMA-3B. Even on larger models like LLaMA2-7B, Ferret maintains a comparable convergence speed to FedAvg, which is still considerably faster than FedKSeed.

Overall, these results highlight the scalability of Ferret, as discussed in Sec. 4.3, and demonstrate its ability to balance computational efficiency, communication overhead, and fast convergence.

## 6  CONCLUSION

In conclusion, our Ferret algorithm offers a highly desirable solution for the scalable, full-parameter tuning of LLMs in federated environments. By achieving high computational efficiency, fast convergence, and reduced communication overhead, Ferret overcomes the limitations of existing methods, striking an improved balance among these critical factors. Moreover, our rigorous theoretical analyses and extensive experiments validate Ferret as a robust and reliable approach for deploying LLMs in large-scale federated settings.

## REPRODUCIBILITY STATEMENT

Of note, due to limited space, we have provided the related work section in Appx. A. To ensure the reproducibility of the theoretical analysis presented in this paper, we have included complete proofs of all theorems and propositions in Appx. B. Additionally, we provide detailed descriptions of the experimental settings and comprehensive ablation studies in Appx. C.

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

## APPENDIX A  RELATED WORK

**Federated PEFT for LLMs.** The field of federated learning (FL) has gained significant traction in its application to the fine-tuning of large language models (LLMs). Traditional FL approaches in this domain (Zhang et al., 2023; Kuang et al., 2024; Zhang et al., 2024a; Kuang et al., 2023) have predominantly focused on parameter-efficient fine-tuning (PEFT) techniques (Hu et al., 2022; Wei et al., 2024; Lester et al., 2021; Lin et al., 2024; Hu et al., 2024), which reduce the number of trainable parameters in LLMs to mitigate the extensive communication overheads in FL scenarios. Unfortunately, while PEFT methods such as those proposed in (Kuang et al., 2024; Zhang et al., 2024a) have shown promise, they often fall short in achieving the accuracy levels possible with full-parameter tuning (Pu et al., 2023), particularly in non-IID (non-independent and identically distributed) data settings commonly encountered in FL. In contrast, this paper focuses on federated full-parameter tuning of LLMs, aiming to achieve significantly reduced communication overhead while maintaining competitive model accuracy.

**Federated Learning with Shared Randomness.** Several approaches leveraging shared randomness have been proposed to enhance communication efficiency in FL. Methods including (Qin et al., 2024; Xu et al., 2023; Maritan et al., 2023; Feng et al., 2023; Dorfman et al., 2023; Zelikman et al., 2023; Rahimi et al., 2024) demonstrate that by transmitting only a limited set of random seeds and scalar gradients, communication overhead can be drastically reduced. However, these methods rely on zeroth-order optimization (ZOO) for their local updates on each client. This reliance often results in poor scalability, as these methods require substantial computational costs per round to achieve the same local update progress and a larger number of communication rounds to converge compared with their first-order counterparts, such as FedAvg (McMahan et al., 2017) and FedProx (Li et al., 2020). This limitation therefore becomes a bottleneck in large-scale federated environments. In contrast, our paper introduces the use of shared randomness within first-order FL, aiming to improve both computational and communication-round efficiency of zeroth-order FL. To the best of our knowledge, this is the first time that shared randomness has been introduced in first-order FL to reduce communication overhead.

## APPENDIX B  PROOFS

### B.1  PROOF OF THM. 1

Suppose v is randomly and independently sampled from a truncated normal distribution, i.e., $v \sim \mathcal{N}(0,1)$ with $v \in [-1/\sqrt{d}, 1/\sqrt{d}]$, we have

$$\mathbb{E}[v] = 0 , \tag{9}$$

and also

$$\mathbb{E}\left[v^2\right] = \left(\mathbb{E}[v]\right)^2 + \text{VAR}(v)$$

$$= \text{VAR}(v)$$

$$= 1 - \frac{1/\sqrt{d}(\psi(1/\sqrt{d}) + \psi(-1/\sqrt{d}))}{\Phi(1/\sqrt{d}) - \Phi(-1/\sqrt{d})} - \left(\frac{\psi(1/\sqrt{d}) - \psi(-1/\sqrt{d})}{\Phi(1/\sqrt{d}) - \Phi(-1/\sqrt{d})}\right)^2 \tag{10}$$

$$= 1 - \frac{2\psi(1/\sqrt{d})/\sqrt{d}}{2\Phi(1/\sqrt{d}) - 1}$$

where $\psi(\frac{1}{\sqrt{d}})$ and $\Phi(\frac{1}{\sqrt{d}})$ is the probability density function (PDF) and cumulative distribution function (CDF) of the standard normal distribution evaluated at $1/\sqrt{d}$, respectively.

According to Sec. 3.2, each element v in $\mathbf{V}$ is randomly and independently sampled from the truncated normal distribution above. We therefore have the following to conclude our proof:

$$\mathbb{E}\left[\widetilde{\Delta}\right] = \frac{1}{\rho K}\mathbb{E}\left[\mathbf{V}\mathbf{V}^\top\right]\Delta$$

$$= \frac{1}{\rho K}\mathbb{E}\left[\sum_{k=1}^K \mathbf{v}_k\mathbf{v}_k^\top\right]\Delta \tag{11}$$

$$= \Delta \ .$$

### B.2 PROOF OF THM. 2

To begin with, we introduce the lemma below to ease our proof.

**Lemma 1** (Matrix Bernstein Inequality, Thm. 1.6.2 in (Tropp et al., 2015)). *Let* $\mathbf{X}_1, \cdots, \mathbf{X}_K$ *be independent, zero mean, and symmetry matrices of size* $d \times d$, *if* $\|\mathbf{X}_k\| \le C$ *for any* $k \in [K]$, *we then have*

$$\mathbb{E}\left[\left\|\sum_{k=1}^K \mathbf{X}_k\right\|\right] \le \sqrt{2\nu\ln(2d)} + \frac{1}{3}C\ln(2d) \tag{12}$$

*where* $\nu \triangleq \left\|\sum_{k=1}^K \mathbb{E}\left[\mathbf{X}_k^2\right]\right\|$.

Define $\mathbf{X}_k \triangleq \left(\mathbf{v}_k\mathbf{v}_k^\top - \rho\mathbf{I}_d\right)/K$, We have

$$\begin{aligned}
\|\mathbf{X}_k\| &\overset{(a)}{=} \frac{1}{K}\left\|\mathbf{v}_k\mathbf{v}_k^\top - \rho\mathbf{I}_d\right\| \\
&\overset{(b)}{\le} \frac{1}{K}\left(\left\|\mathbf{v}_k\mathbf{v}_k^\top\right\| + \rho\left\|\mathbf{I}_d\right\|\right) \\
&\overset{(c)}{=} \frac{1}{K}\left(\left\|\mathbf{v}_k^\top\mathbf{v}_k\right\| + \rho\right) \\
&\overset{(d)}{\le} \frac{2}{K}
\end{aligned} \tag{13}$$

where $(b)$ comes from triangle inequality and $(c)$ is due to the fact that outer product $\mathbf{v}_k\mathbf{v}_k^\top$ and inner product $\mathbf{v}_k^\top\mathbf{v}_k$ shares the same operator norm. Finally, $(d)$ results from $\rho < 1$ and $\mathbf{v}_k^\top\mathbf{v}_k \le 1$.

Besides, we also have

$$\begin{aligned}
\mathbb{E}\left[\mathbf{X}_k^2\right] &\overset{(a)}{=} \frac{1}{K^2}\mathbb{E}\left[\mathbf{v}_k\mathbf{v}_k^\top\mathbf{v}_k\mathbf{v}_k^\top - 2\rho\mathbf{v}_k\mathbf{v}_k^\top + \rho^2\mathbf{I}_d\right] \\
&\overset{(b)}{\preceq} \frac{1}{K^2}\mathbb{E}\left[\mathbf{v}_k\mathbf{v}_k^\top - 2\rho\mathbf{v}_k\mathbf{v}_k^\top + \rho^2\mathbf{I}_d\right] \\
&\overset{(c)}{=} \frac{1}{K^2}\left(\rho - \rho^2\right)\mathbf{I}_d \\
&\overset{(d)}{\preceq} \frac{\rho}{K^2}\mathbf{I}_d
\end{aligned} \tag{14}$$

where $(b)$ comes from the fact that $\mathbf{v}_k^\top\mathbf{v}_k \le 1$ and $(c)$ is due to the fact that $\mathbb{E}\left[\mathbf{v}_k\mathbf{v}_k^\top\right] = \rho\mathbf{I}_d$.

As a result, by introducing the results above with a triangle inequality, we have

$$\left\|\sum_{k=1}^K \mathbb{E}\left[\mathbf{X}_k^2\right]\right\| \le \sum_{k=1}^K \left\|\frac{\rho}{K^2}\mathbf{I}_d\right\| \tag{15}$$

$$\le \frac{\rho}{K} \ .$$

By introducing the results above into Lemma. 1,

$$
\begin{aligned}
\mathbb{E}\left[\left\|\widetilde{\Delta} - \Delta\right\|\right] &= \mathbb{E}\left[\left\|\frac{1}{\rho K}\mathbf{V}\mathbf{V}^\top\Delta - \Delta\right\|\right] \\
&\leq \mathbb{E}\left[\left\|\frac{1}{\rho K}\mathbf{V}\mathbf{V}^\top - \mathbf{I}_d\right\|\right]\|\Delta\| \\
&= \frac{1}{\rho}\mathbb{E}\left[\left\|\sum_{k=1}^K \mathbf{X}_k\right\|\right]\|\Delta\| \\
&\leq \sqrt{\frac{2\ln(2d)}{\rho K}} + \frac{\ln(2d)}{\rho K} \;,
\end{aligned}
\tag{16}
$$

which finally concludes our proof.

**Remark 1.** Sampling from a truncated normal distribution (rather than a standard normal distribution) ensures a bounded norm, which is crucial for achieving a bounded reconstruction error by our method in Sec. 3.2.

### B.3 PROOF OF THM. 3

Since the loss function $\ell(\cdot; \cdot)$ is assumed to be $\beta$-smooth w.r.t its first argument, we then have

$$
\begin{aligned}
\ell(\mathbf{w} + \epsilon\mathbf{v}_k; \mathbf{x}^{(i)}) - \ell(\mathbf{w}; \mathbf{x}^{(i)}) &\leq \epsilon\left(\nabla\ell(\mathbf{w}; \mathbf{x}^{(i)})\right)^\top \mathbf{v}_k + \frac{1}{2}\beta\epsilon^2\|\mathbf{v}_k\|^2 \\
&\leq \epsilon\left(\nabla\ell(\mathbf{w}; \mathbf{x}^{(i)})\right)^\top \mathbf{v}_k + \frac{1}{2}\beta\epsilon^2 \;.
\end{aligned}
\tag{17}
$$

By dividing $\epsilon$ on both sides of the inequality above, we have

$$
g_k - \mathbf{v}_k^\top \nabla\ell(\mathbf{w}; \mathbf{x}^{(i)}) \leq \frac{1}{2}\beta\epsilon \;.
\tag{18}
$$

We therefore can conclude our proof using the results below:

$$
\begin{aligned}
\left\|\frac{1}{K}\mathbf{V}\mathbf{g} - \frac{1}{K}\mathbf{V}\mathbf{V}^\top\nabla\ell(\mathbf{w}; \mathbf{x}^{(i)})\right\| &\overset{(a)}{=} \left\|\frac{1}{K}\sum_{k=1}^K\left(\mathbf{v}_k g_k - \mathbf{v}_k\mathbf{v}_k^\top\nabla\ell(\mathbf{w}; \mathbf{x}^{(i)})\right)\right\| \\
&\overset{(b)}{\leq} \frac{1}{K}\sum_{k=1}^K\left\|\mathbf{v}_k g_k - \mathbf{v}_k\mathbf{v}_k^\top\nabla\ell(\mathbf{w}; \mathbf{x}^{(i)})\right\| \\
&\overset{(c)}{\leq} \frac{1}{K}\sum_{k=1}^K\left|g_k - \mathbf{v}_k^\top\nabla\ell(\mathbf{w}; \mathbf{x}^{(i)})\right|\|\mathbf{v}_k\| \\
&\overset{(d)}{\leq} \frac{1}{2}\beta\epsilon
\end{aligned}
\tag{19}
$$

where $(b)$ comes from triangle inequality and $(d)$ results from (18) and $\|\mathbf{v}_k\| \leq 1$.

**Remark 2.** When $\epsilon \to 0$, (18) indicates that $g_k = \mathbf{v}_k^\top\nabla\ell(\mathbf{w}; \mathbf{x}^{(i)})$, implying that this scalar gradient in zeroth-order method, e.g., FedKSeed (Qin et al., 2024), is an approximation of directional derivative, i.e., our projected update in (6) when $\Delta$ is replaced with $\nabla\ell(\mathbf{w}; \mathbf{x}^{(i)})$.

### B.4  PROOF OF PROP. 1

Due to the fact that $d = \sum_{l \in [L]} d_l$, $K = \sum_{l \in [L]} K_l$, and $K_l > 0$ for any $l \in [L]$, we have

$$
\begin{aligned}
dK &= \left( \sum_{l \in [L]} d_l \right) \left( \sum_{l \in [L]} K_l \right) \\
&= \sum_{l \in [L]} d_l \left( \sum_{l \in [L]} K_l \right) \\
&> \sum_{l \in [L]} d_l K_l \, ,
\end{aligned}
$$

which therefore concludes our proof.

### B.5  PROOF OF PROP. 2

Based on our block-wise reconstruction in (8) and Thm. 2, we have

$$
\begin{aligned}
\mathbb{E}\left[ \left\| \widetilde{\Delta} - \Delta \right\| \right] &\overset{(a)}{=} \mathbb{E}\left[ \sqrt{ \sum_{l \in [L]} \left\| \widetilde{\Delta}_l - \Delta_l \right\|^2 } \right] \\
&\overset{(b)}{<} \mathbb{E}\left[ \sqrt{ \left( \sum_{l \in [L]} \left\| \widetilde{\Delta}_l - \Delta_l \right\| \right)^2 } \right] \\
&\overset{(c)}{=} \sum_{l \in [L]} \mathbb{E}\left[ \left\| \widetilde{\Delta}_l - \Delta_l \right\| \right] \\
&\overset{(d)}{\le} \sum_{l \in [L]} \left( \sqrt{ \frac{2 \ln(2 d_l)}{\rho_l K_l} } + \frac{\ln(2 d_l)}{\rho_l K_l} \right) \|\Delta_l\|
\end{aligned}
\tag{20}
$$

where $(a)$ is based on the definition of $\widetilde{\Delta}_l$ and $\Delta_l$ and $(b)$ is from the fact that $\left\| \widetilde{\Delta}_l - \Delta_l \right\| > 0$.

Given that $\sqrt{d_l} > K_l$ and we can then use $\widetilde{\mathcal{O}}$ to hide the logarithm term in the result above, the following then holds:

$$
\mathbb{E}\left[ \left\| \widetilde{\Delta} - \Delta \right\| \right] < \widetilde{\mathcal{O}}\left( \sum_{l \in [L]} \frac{\|\Delta_l\|}{\rho_l K_l} \right) \, .
\tag{21}
$$

To minimize the upper bound above w.r.t $\{K_l\}_{l=1}^L$ with $\sum_{l \in [L]} K_l = K$, we resort to KKT conditions. Specifically, define $\boldsymbol{k} \triangleq [K_1, \cdots, K_L]^\top$ and the following Lagrangian function based on $\lambda > 0$:

$$
F(\boldsymbol{k}, \lambda) \triangleq \sum_{l \in [L]} \frac{\|\Delta_l\|}{\rho_l K_l} + \lambda \left( \sum_{l \in [L]} K_l - K \right) \, .
\tag{22}
$$

To minimize (21), for any $l \in [L]$, $K_l$ and $\lambda$ then needs to satisfy the following condition:

$$
\frac{\partial F(\boldsymbol{k}, \lambda)}{\partial K_l} = - \frac{\|\Delta_l\| / \rho_l}{K_l^2} + \lambda = 0 \, .
\tag{23}
$$

That is,

$$
\lambda = \frac{\|\Delta_1\| / \rho_1}{K_1^2} = \cdots = \frac{\|\Delta_L\| / \rho_L}{K_L^2} \, .
\tag{24}
$$

This finally leads to $K_l \propto \sqrt{\|\Delta_L\| / \rho_L}$, which consequently concludes our proof.

**Remark 3.** Prop. 2 provides a looser bound than Thm. 2, primarily owing to the inequality $(b)$ in (20). Based on this looser bound, one might expect that block-wise reconstruction would incur a larger error compared to the vanilla reconstruction in (7). However, empirical results in Appx. C.4 and Appx. C.5 show that block-wise reconstruction yields comparable performance to the vanilla approach.

B.6  PROOF OF THM. 4

Of note, we follow the general idea in (Shu et al., 2024) to prove the convergence of Ferret. To begin with, we introduce the following lemmas borrowed from (Shu et al., 2024):

**Lemma 2.** *Let $\{\boldsymbol{u}_1,\ldots,\boldsymbol{u}_\tau\}$ be any $\tau$ vectors in $\mathbb{R}^d$. Then the following holds for any $a > 0$:*

$$\|\boldsymbol{u}_i\|\,\|\boldsymbol{u}_j\| \le \frac{a}{2}\|\boldsymbol{u}_i\|^2 + \frac{1}{2a}\|\boldsymbol{u}_j\|^2 \ , \tag{25}$$

$$\|\boldsymbol{u}_i + \boldsymbol{u}_j\|^2 \le (1+a)\|\boldsymbol{u}_i\|^2 + \left(1 + \frac{1}{a}\right)\|\boldsymbol{u}_j\|^2 \ , \tag{26}$$

$$\left\|\sum_{i=1}^\tau \boldsymbol{u}_i\right\|^2 \le \tau \sum_{i=1}^\tau \|\boldsymbol{u}_i\|^2 \ . \tag{27}$$

**Lemma 3.** *For any $\beta$-smooth function $f$, inputs $\boldsymbol{x},\boldsymbol{y}$ in the domain of $f$, the following holds for any constant $\eta > 0$:*

$$\|\boldsymbol{x} - \eta\nabla f(\boldsymbol{x}) - \boldsymbol{y} + \eta\nabla f(\boldsymbol{y})\|^2 \le (1+\eta\beta)^2\|\boldsymbol{x}-\boldsymbol{y}\|^2 \ .$$

Let $\eta \le 1/(T\beta)$, we can bound the discrepancy between $\mathbf{w}_{r,t}^{(i)}$ and $\mathbf{w}_r$ for any client $i$ as below

$$
\begin{aligned}
&\mathbb{E}\left[\left\|\mathbf{w}_{r,t}^{(i)} - \mathbf{w}_r\right\|^2\right]\\
&\overset{(a)}{=} \mathbb{E}\left[\left\|\mathbf{w}_{r,t-1}^{(i)} - \eta\nabla\ell(\mathbf{w}_{r,t-1}^{(i)};\mathbf{x}_{r,t-1}^{(i)}) - \mathbf{w}_r\right\|^2\right]\\
&\overset{(b)}{=} \mathbb{E}\left[\left\|\mathbf{w}_{r,t-1}^{(i)} - \eta\nabla\mathcal{L}(\mathbf{w}_{r,t-1}^{(i)}) + \eta\nabla\mathcal{L}(\mathbf{w}_r) - \mathbf{w}_r + \eta\left(\nabla\mathcal{L}(\mathbf{w}_{r,t-1}^{(i)}) - \nabla\mathcal{L}^{(i)}(\mathbf{w}_{r,t-1}^{(i)})\right)\right.\right.\\
&\qquad\qquad \left.\left.+\eta\left(\nabla\mathcal{L}^{(i)}(\mathbf{w}_{r,t-1}^{(i)}) - \nabla\ell(\mathbf{w}_{r,t-1}^{(i)};\mathbf{x}_{r,t-1}^{(i)}) - \nabla\mathcal{L}(\mathbf{w}_r)\right)\right\|^2\right]\\
&\overset{(c)}{\le} \frac{T}{T-1}\mathbb{E}\left[\left\|\mathbf{w}_{r,t-1}^{(i)} - \eta\nabla\mathcal{L}(\mathbf{w}_{r,t-1}^{(i)}) + \eta\nabla\mathcal{L}(\mathbf{w}_r) - \mathbf{w}_r\right\|^2\right]\\
&\qquad + 2\eta^2 T\,\mathbb{E}\left[\left\|\nabla\mathcal{L}^{(i)}(\mathbf{w}_{r,t-1}^{(i)}) - \nabla\ell(\mathbf{w}_{r,t-1}^{(i)};\mathbf{x}_{r,t-1}^{(i)})\right\|^2 + \|\nabla\mathcal{L}(\mathbf{w}_r)\|^2\right]\\
&\overset{(d)}{\le} \frac{T(1+\eta\beta)^2}{T-1}\mathbb{E}\left[\left\|\mathbf{w}_{r,t-1}^{(i)} - \mathbf{w}_r\right\|^2\right] + 2\eta^2 T\sigma^2 + 2\eta^2 T\|\nabla\mathcal{L}(\mathbf{w}_r)\|^2\\
&\overset{(e)}{\le} 24\eta^2 T^2\sigma^2 + 24\eta^2 T^2\|\nabla\mathcal{L}(\mathbf{w}_r)\|^2
\end{aligned}
\tag{28}
$$

where $(a)$ is from the local update of $\mathbf{w}_{r,t-1}^{(i)}$ on each client $i$, and $(c)$ is based on (26) in Lemma 2 with $a = 1/(T-1)$ and $\mathcal{L}(\mathbf{w}_{r,t-1}^{(i)}) = \mathcal{L}^{(i)}(\mathbf{w}_{r,t-1}^{(i)})$. Besides, $(d)$ results from Lemma 3 and the assumption that $\mathbb{E}[\|\nabla\mathcal{L}^{(i)}(\mathbf{w}) - \ell(\mathbf{w};\mathbf{x})\|^2] \le \sigma^2$. Finally, $(e)$ comes from the summation of

geometric series and the fact that $\eta\beta \leq 1/T$ as well as

$$
\begin{aligned}
\sum_{\tau=0}^{t-1}\left(\frac{(T+1)^2}{T(T-1)}\right)^\tau &\leq \sum_{\tau=0}^{T-1}\left(\frac{(T+1)^2}{T(T-1)}\right)^\tau \\
&= \frac{\left((T+1)^2/[T(T-1)]\right)^T - 1}{(T+1)^2/[T(T-1)] - 1} \\
&= \frac{T(T-1)}{3T+1}\left(\left(1+\frac{3T+1}{T(T-1)}\right)^T - 1\right) \\
&< \frac{T(T-1)}{3T+1}\left(\exp\left(\frac{3T+1}{T}\right) - 1\right) \\
&< \frac{T}{3}\left(\exp\left(\frac{7}{2}\right) - 1\right) \\
&< 12T \, .
\end{aligned}
\tag{29}
$$

Besides $\mathbb{E}\left[\mathbf{V}_r\mathbf{V}_r^\top\right] = \rho\mathbf{I}_d$, one can also verify that $\mathbb{E}\left[\mathbf{V}_r\mathbf{V}_r^\top\mathbf{V}_r\mathbf{V}_r^\top\right] = \rho^2\mathbf{I}_d$, we therefore have

$$
\begin{aligned}
&\mathbb{E}\left[\|\mathbf{w}_{r+1} - \mathbf{w}_r\|^2\right] \\
&= \mathbb{E}\left[(\mathbf{w}_{r+1} - \mathbf{w}_r)^\top(\mathbf{w}_{r+1} - \mathbf{w}_r)\right] \\
&= \mathbb{E}\left[\left(\frac{\eta}{\rho KN}\right)^2\left(\sum_{i=1}^{N}\sum_{t=1}^{T}\nabla\ell(\mathbf{w}_{r,t-1}^{(i)};\mathbf{x}_{r,t-1}^{(i)})\right)^\top \mathbf{V}_r\mathbf{V}_r^\top\mathbf{V}_r\mathbf{V}_r^\top \sum_{i=1}^{N}\sum_{t=1}^{T}\nabla\ell(\mathbf{w}_{r,t-1}^{(i)};\mathbf{x}_{r,t-1}^{(i)})\right] \\
&= \left(\frac{\eta}{\rho KN}\right)^2 \mathbb{E}\left[\left(\sum_{i=1}^{N}\sum_{t=1}^{T}\nabla\ell(\mathbf{w}_{r,t-1}^{(i)};\mathbf{x}_{r,t-1}^{(i)})\right)^\top \mathbb{E}\left[\mathbf{V}_r\mathbf{V}_r^\top\mathbf{V}_r\mathbf{V}_r^\top\right]\sum_{i=1}^{N}\sum_{t=1}^{T}\nabla\ell(\mathbf{w}_{r,t-1}^{(i)};\mathbf{x}_{r,t-1}^{(i)})\right] \\
&= \left(\frac{\eta}{N}\right)^2 \mathbb{E}\left[\left\|\sum_{i=1}^{N}\sum_{t=1}^{T}\nabla\ell(\mathbf{w}_{r,t-1}^{(i)};\mathbf{x}_{r,t-1}^{(i)})\right\|^2\right] \\
&\leq \frac{\eta^2}{N}\sum_{i=1}^{N}\mathbb{E}\left[\left\|\sum_{t=1}^{T}\nabla\ell(\mathbf{w}_{r,t-1}^{(i)};\mathbf{x}_{r,t-1}^{(i)})\right\|^2\right]
\end{aligned}
\tag{30}
$$

where the last inequality comes from the (27) in Lemma 2. Here, we omit the subscript $r$ from the random bases $\mathbf{V}$ in our notation for simplicity.

Since $\mathbb{E}\left[\left\|\mathbf{w}_{r,t}^{(i)} - \mathbf{w}_r\right\|^2\right] = \eta^2\mathbb{E}\left[\left\|\sum_{\tau=1}^{t}\nabla\ell(\mathbf{w}_{r,\tau-1}^{(i)};\mathbf{x}_{r,\tau-1}^{(i)})\right\|^2\right]$, by replacing $\tau$ with $T$, we have

$$
\mathbb{E}\left[\|\mathbf{w}_{r+1} - \mathbf{w}_r\|^2\right] \leq 24\eta^2 T^2\sigma^2 + 24\eta^2 T^2\|\nabla\mathcal{L}(\mathbf{w}_r)\|^2 \, .
\tag{31}
$$

Besides, since $\mathbb{E}\left[\mathbf{w}_{r+1} - \mathbf{w}_r\right] = -\frac{\eta}{N} \sum_{i=1}^{N} \sum_{t=1}^{T} \nabla\mathcal{L}^{(i)}(\mathbf{w}_{r,t-1}^{(i)})$, we have

$$
\mathbb{E}\left[\nabla\mathcal{L}(\mathbf{w}_r)^\top (\mathbf{w}_{r+1} - \mathbf{w}_r)\right]
$$

$$
\overset{(a)}{=} -\frac{\eta}{N}\mathbb{E}\left[\sum_{i=1}^{N}\sum_{t=1}^{T} \nabla\mathcal{L}(\mathbf{w}_r)^\top \nabla\mathcal{L}^{(i)}(\mathbf{w}_{r,t-1}^{(i)})\right]
$$

$$
\overset{(b)}{=} -\frac{\eta}{N}\mathbb{E}\left[\sum_{i=1}^{N}\sum_{t=1}^{T} \nabla\mathcal{L}(\mathbf{w}_r)^\top \left(\nabla\mathcal{L}^{(i)}(\mathbf{w}_{r,t-1}^{(i)}) - \nabla\mathcal{L}(\mathbf{w}_{r,t-1}^{(i)}) + \nabla\mathcal{L}(\mathbf{w}_{r,t-1}^{(i)}) - \nabla\mathcal{L}(\mathbf{w}_r) + \nabla\mathcal{L}(\mathbf{w}_r)\right)\right]
$$

$$
\overset{(c)}{\leq} \frac{\eta}{N}\sum_{i=1}^{N}\sum_{t=1}^{T}\left(\eta\beta T\left\|\nabla\mathcal{L}(\mathbf{w}_r)\right\|^2 + \frac{\beta}{4\eta T}\mathbb{E}\left[\left\|\mathbf{w}_{r,t-1}^{(i)} - \mathbf{w}_r\right\|^2\right]\right) - \eta T\left\|\nabla\mathcal{L}(\mathbf{w}_r)\right\|^2
$$

$$
\overset{(d)}{\leq} \left(7\eta^2 T^2\beta - \eta T\right)\left\|\nabla\mathcal{L}(\mathbf{w}_r)\right\|^2 + 6\eta^2 T^2\beta\sigma^2
$$

$$\tag{32}$$

where $(c)$ comes from Cauchy–Schwarz inequality and the fact that $\mathcal{L}(\mathbf{w}_{r,t-1}^{(i)}) = \mathcal{L}^{(i)}(\mathbf{w}_{r,t-1}^{(i)})$. In addition, $(d)$ results from (31).

Finally, based on the assumption that $\mathcal{L}$ is $\beta$-smooth, we naturally have

$$
\mathbb{E}\left[\mathcal{L}(\mathbf{w}_{r+1}) - \mathcal{L}(\mathbf{w}_r)\right] \leq \mathbb{E}\left[\nabla\mathcal{L}(\mathbf{w}_r)^\top (\mathbf{w}_{r+1} - \mathbf{w}_r)\right] + \frac{\beta}{2}\mathbb{E}\left[\left\|\mathbf{w}_{r+1} - \mathbf{w}_r\right\|^2\right]
$$

$$
\leq (19\eta^2 T^2\beta - \eta T)\mathbb{E}\left[\left\|\nabla\mathcal{L}(\mathbf{w}_r)\right\|^2\right] + 18\eta^2 T^2\beta\sigma^2 \ .
$$

$$\tag{33}$$

By rearranging and letting $\eta \leq \frac{1}{20T\beta}$, we have

$$
\mathbb{E}\left[\left\|\nabla\mathcal{L}(\mathbf{w}_r)\right\|^2\right] \leq \frac{20\,\mathbb{E}\left[\mathcal{L}(\mathbf{w}_r) - \mathcal{L}(\mathbf{w}_{r+1})\right]}{\eta T} + 360\eta T\beta\sigma^2 \ ,
$$

$$\tag{34}$$

Finally, by summarizing both sides over $R$ rounds and scaling them with $1/R$, we have the following results to conclude our proof:

$$
\min_{r \in [R)} \mathbb{E}\left[\left\|\nabla\mathcal{L}(\mathbf{w}_r)\right\|^2\right] \leq \frac{20\left(\mathcal{L}(\mathbf{w}_0) - \min_{\mathbf{w}}\mathcal{L}(\mathbf{w})\right)}{\eta T R} + 360\eta\beta T\sigma^2 \ .
$$

$$\tag{35}$$

**Remark 4.** Note that the large constant in (35) arises from our bound in (28) for sufficiently large $T$. This bound can be improved in practice by considering a smaller $T$ instead.

# APPENDIX C EXPERIMENTS

## C.1 EXPERIMENTAL SETUP

**Baselines.** In line with the comparison in (Qin et al., 2024), we selected four practical methods for federated LLM tuning as our baselines: (1) FedPTuning (Kuang et al., 2024), (2) FedPrompt (Kuang et al., 2023), (3) FedIT (Zhang et al., 2024a), and (4) FedIT-SGD, a variant of FedIT that replaces Adam with SGD. In addition, we included four full-parameter tuning methods for comparison: (1) FedAvg (McMahan et al., 2017), (2) FedZO (Fang et al., 2022), (3) FedMeZO, a hybrid of FedAvg and MeZO (Malladi et al., 2023), and (4) FedKSeed (Qin et al., 2024).

### C.1.1 SETUP ON THE NATURAL INSTRUCTION AND DOLLY-15K DATASETS

**Datasets.** We conducted our experiments using the Natural Instructions (NI) (Wang et al., 2022) and Dolly-15K (Conover et al., 2023) datasets, following a setup similar to (Qin et al., 2024). For the NI dataset, we allocated 738 training tasks to individual clients for local updates and reserved 119 test tasks for global evaluation, reflecting a non-IID distribution. Meanwhile, for the Dolly-15K dataset, the final task was utilized for global evaluation, while the remaining tasks were distributed among 200 clients with varying levels of label distribution skew. Rouge-L (Lin, 2004) was chosen as the evaluation metric. Given our resource constraints, we selected DataJuicer-1.3B (Chen et al., 2023) and LLaMA-3B (Touvron et al., 2023a) as the base models for our study. The corresponding HuggingFace model paths are "datajuicer/LLaMA-1B-dj-refine-150B" and "openlm-research/open_llama_3b".

**FL Settings.** In each round of federated learning, 5% of clients were randomly selected to participate. Following the same practice in FedKSeed (Qin et al., 2024), we set the total number of communication rounds to 40 for the NI dataset and 60 for Dolly-15K for all baselines. Due to the compelling efficiency of our method, we set the total number of communication rounds to 12 for the NI dataset and 20 for Dolly-15K for Ferret. First-order baselines trained locally for one epoch, and FedKSeed trained for 200 steps, while our Ferret algorithm trained for 10 iterations (i.e., $T = 10$ in Algo. 1). The $K$ value was set to 4096 for FedKSeed. All approaches perform local update with a batchsize of 1 to reduce memory consumption. For each local update iteration in Ferret, we accumulate the gradients from 4 samples.

**Hyper-parameters.** For Ferret, the local update learning rate $\eta$ for each client is set to $1 \times 10^{-4}$, where the selected learning rate is searched from $[2 \times 10^{-4}, 1 \times 10^{-4}, 5 \times 10^{-5}]$. The global aggregation learning rates on Natural Instruction and Dolly-15K are set to $10.0$ and $3.0$, respectively, which is search from $[10.0, 5.0, 1.0]$. For other baselines in Tab. 1 of our main paper, we reported their accuracy performances using the results from FedKSeed (Qin et al., 2024).

**Prompt Template.** In our experiments, the raw input data is pre-processed to follow a structured format, where we warp the input text to the Alpaca prompt template (Taori et al., 2023). The corresponding templates for the NI and Dolly-15K dataset are shown in Table 6 and 7.

Table 6: Prompt template for Natural Instructions.

Below is an instruction that describes a task, paired with an input that provides further context. Write a response that appropriately completes the request.

### Instruction: {Definition}

### Input: {input}

### Response:

Table 7: Prompt template for Dolly-15K. If some data instances do not have the `context` attribute, we will discard the line "### Input: " in the template.

---

Below is an instruction that describes a task, paired with an input that provides further context. Write a response that appropriately completes the request.

### Instruction: {instruction}

### Input: {context}

### Response:

---

### C.1.2    SETUP ON THE CODEALPACA AND GSM8K DATASETS

**Datasets.** To further demonstrate that Ferret can also improve the capability of larger LLMs for code generation and mathematical reasoning, we conducted more experiments using the CodeAlpaca (Chaudhary, 2023) and GSM8K (Cobbe et al., 2021) datasets, following a similar federated setup. The CodeAlpaca dataset (of around 8.0k samples) is a code dataset that consists of ten programming languages, including C, C#, C++, Go, Java, PHP, Pascal, Python, Scale, and X86-64 Assemble. We exclude the X86-64 Assembly data due to limited samples in the dataset. We uniformly randomly sampled 10% instances from the original data as the hold-out test set for evaluation, and we split the remaining 10% samples into nine subsets based on the programming language category and assign each subset to one client as its local training data. For GSM8K, its official train set is split into three subsets, where each client's dataset consists of grade school math questions randomly partitioned from the original dataset, forming a IID distribution. We use the official GSM8K test split as the evaluation dataset. Rouge-L (Lin, 2004) was chosen as the evaluation metric. To demonstrate the scalability of Ferret, we extended the experiments to larger models: LLaMA2-7B and LLaMA2-13B (Touvron et al., 2023a) as the base models for our study. The corresponding HuggingFace model paths are "meta-llama/Llama-2-7b-hf" and "meta-llama/Llama-2-13b-hf".

**FL Settings.** Due to the computing constraints, we set the total number of communication rounds to 20 for both CodeAlpaca and GSM8K for all methods. The $K$ value was set to 4096 for FedKSeed as the same as before. Zeroth-order baselines are trained locally for 200 steps, while FedAvg and Ferret are trained for 10 iterations with accumulating the gradients from 4 samples. All approaches perform local updates with a batch size of 1 to reduce memory consumption.

**Hyper-parameters.** For FedZO and FedKSeed, the local update learning rate is set to $3 \times 10^{-7}$ for all models. For FedAvg on both LLaMA2-7B and LLaMA2-13B, the local update learning rate $\eta$ for each client is set to $3 \times 10^{-4}$, and the global aggregation learning rate is set to $1.0$. For Ferret on LLaMA2-7B, the local update learning rate $\eta$ is set to $3 \times 10^{-4}$ and the global aggregation learning rate is set to $5.0$. For Ferret on LLaMA2-13B, the local update learning rate $\eta$ is set to $5 \times 10^{-4}$ and the global aggregation learning rate is set to $10.0$. The selected learning rate is searched from $[5 \times 10^{-4}, 3 \times 10^{-4}, 1 \times 10^{-4}]$ and the selected global aggregation learning rates is searched from $[10.0, 5.0, 1.0]$.

### C.2    CALCULATION OF COMPUTATIONAL COST AND COMMUNICATION OVERHEAD

In this subsection, we provide the details of how the computational cost and communication cost are calculated for all methods listed in Tab. 4.

**Calculation of Computational Cost.** For FedZO, we follow the same hyper-parameters ($b_1 = 200, b_2 = 1$) for FedZO from FedKSeed paper (Qin et al., 2024), which employs 200 local update steps and 1 perturbation for each local update step. For calculating the computational cost of FedAvg and Ferret, we apply 10 local update steps for each client. Same as our experimental setting in Tab. 1,

Table 8: Comparison of computational cost and communication overhead on LLaMA2-13B, focusing on (a) the computational costs from local updates, global aggregation, and the overall tuning process; and (b) the per-round and overall communication costs. The improvement achieved by our Ferret is reported in brackets using blue (compared with FedKSeed) and orange (compared with FedAvg).

| Algorithm | Computational Cost (Sec.) | | | Communication Cost (# param.) | |
|---|---|---|---|---|---|
| | Local Update | Global Aggr. | Overall | Per-Round | Overall |
| FedZO | 114.1 | 25.7 | $2.8\times10^3$ | $2.6\times10^{10}$ | $5.2\times10^{11}$ |
| FedKSeed | 188.4 | 666.2 | $1.7\times10^4$ | $8.2\times10^3$ | $1.6\times10^5$ |
| FedAvg | 24.9 | 25.7 | $1.0\times10^3$ | $2.6\times10^{10}$ | $5.2\times10^{11}$ |
| Ferret (ours) | **19.2** (9.8×) | **169.4** (3.9×) | $\mathbf{3.8\times10^3}$ (4.5×) | $\mathbf{7.6\times10^3}$ $(10^6\times)$ | $\mathbf{1.5\times10^4}$ $(10^7\times)$ |

the batch size is set to 1 for all methods. The time cost incurred at gradient projection is also included in the Local Update.

In the global aggregation process for both FedZO and FedAvg, raw gradients from all clients are averaged and then used to update the global model. In contrast, for FedKSeed and Ferret, the projected gradients are first aggregated through averaging, then reconstructed, and finally used to update the global model.

For the overall computation cost, we follow the calculation below:

$$\text{Overall} = (\text{Local Update} + \text{Global Aggr.}) \times R.$$

**Calculation of Communication Overhead.** The per-round communication cost refers to the total number of parameters exchanged between a client and the central server during a single round. This includes both the raw or projected gradients that the client sends to the server and the aggregated gradients that the client receives from the server. Each parameter (or projected gradient) is encoded as 16-bit floating point numbers. In accordance with the practice in FedKSeed, we set the number of rounds $R$ to 40 for both FedZO and FedKSeed. Given the notable convergence rate of Ferret, we set $R$ to 12 for both Ferret and FedAvg. Although (Qin et al., 2024) employs $R = 40$ for FedAvg, we use $R = 12$ to provide a strong basis for comparison and to highlight the computational efficiency of Ferret.

For the overall communication cost, we follow the calculation below:

$$\text{Overall} = \text{Per-Round} \times R.$$

### C.3 MORE COMPARISON OF COMPUTATIONAL COST AND COMMUNICATION OVERHEAD

Table 8 compares the computational cost and communication overhead of LLaMA2-13B using the GSM8K dataset. Because of GPU memory constraints, FedZO and FedAvg have slightly higher computational costs, as gradients need to be stored on the CPU. The results show that even for large models like LLaMA2-13B, Ferret still demonstrates superior scalability. Compared to FedKSeed, Ferret reduces computational costs significantly: 9.8× for local updates, 3.9× for global aggregation, and 4.5× for overall tuning cost. Additionally, compared to FedAvg, which does not utilize shared randomness, Ferret achieves a dramatic $10^7\times$ reduction in communication costs. These results, along with the evidence in Sec. 5, further highlight the scalability of Ferret in federated full-parameter tuning.

### C.4 ABLATION STUDIES ON RECONSTRUCTION

**Rate of** $1/\rho$ **w.r.t Dimension** $d$. In Fig. 2, we present the rate of $1/\rho$ where $\rho$ is defined in Sec. 3.2 to verify our claim following Thm. 2. The results in Fig. 2 confirm that $1/\rho$ indeed follows a rate of $\mathcal{O}(d)$.

**Comparison of Reconstruction Accuracy between Ferret and ZO Method.** In Fig. 3, we present the reconstruction accuracy (measured by cosine similarity) for the $d = 10^5$-dimensional gradient of

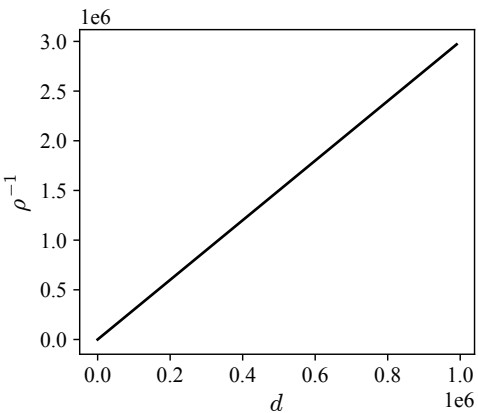

Figure 2: Rate of $1/\rho$ w.r.t. dimension $d$.

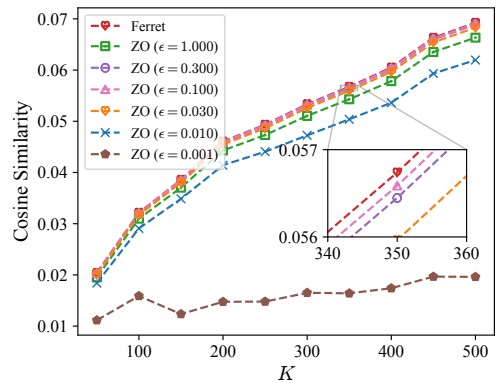

Figure 3: Reconstruction accuracy of our (8) vs. zeroth-order method under varying $K$ and $\epsilon$.

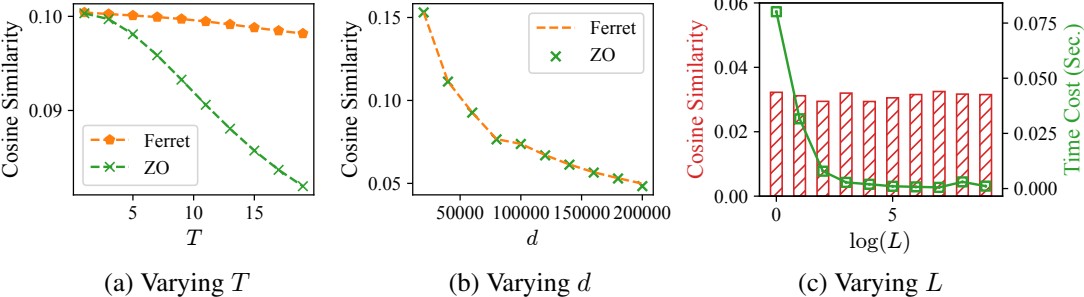

(a) Varying $T$     (b) Varying $d$     (c) Varying $L$

Figure 4: Reconstruction Accuracy (measured by cosine similarity between reconstruction and ground truth) of our (8) vs. zeroth-order method under varying $T$, $d$, and $L$.

the function $F(\boldsymbol{x}) = \sum_{i=1}^{d} x_i^2$ at a randomly sampled input $\boldsymbol{x}$ with varying $K$ by using our method in (7) and zeroth-order method with different values of $\epsilon$. The goal is to compare the reconstruction accuracy of our (7) with that of the ZO method under varying $K$ and $\epsilon$. The results in Fig. 3 indicate that: *(a)* our method (7) achieves improved reconstruction accuracy compared to the ZO method, particularly the one with an optimal $\epsilon = 0.1$, which indeed aligns with the insights from our Thm. 3; *(b)* both our method (7) and the ZO method exhibit the same increasing rate in reconstruction accuracy as $K$ increases, highlighting the connection between these two methods as implied by our Thm. 3; and *(c)* this increasing rate is generally linear, which is consistent with Thm. 2. These results therefore further verify the insights in Thm. 2 and Thm. 3, and support the advantages of our method (7) over the ZO method.

**Reconstruction Accuracy of Ferret under Varying $T$.** In Fig. 4 (a), we present the reconstruction accuracy (measured by cosine similarity) of a $T$-iteration gradient descent update for the function $F(\boldsymbol{x}) = \sum_{i=1}^{d} \sin^2(x_i)$ with a learning rate of 0.1, $d = 5 \times 10^4$, $L = 1$, and $K = 500$, using our method in (8) and the zeroth-order (ZO) method described in Thm. 3 with $\epsilon = 0.1$. The goal is to compare the accumulated error from our (8) with that of the ZO method. Interestingly, Fig. 4 (a) shows that our method maintains consistent reconstruction accuracy as the number $T$ of gradient descent iterations increases, whereas the ZO method experiences a noticeable decline in accuracy. This result implies that our (8) effectively avoids the accumulated error typical in zeroth-order methods, aligning with the theoretical justification provided in Sec. 4.1.

**Reconstruction Accuracy of Ferret under Varying $d$.** In Fig. 4 (b), we show the reconstruction accuracy (measured by cosine similarity) of $d$-dimensional gradient of $F(\boldsymbol{x}) = \sum_{i=1}^{d} \sin^2(x_i)$ at a randomly sampled input $\boldsymbol{x}$, with $L = 1$ and $K = 500$, using our method in (8) and the zeroth-order (ZO) method described in Thm. 3 with $\epsilon = 0.1$. The goal is to compare the reconstruction accuracy

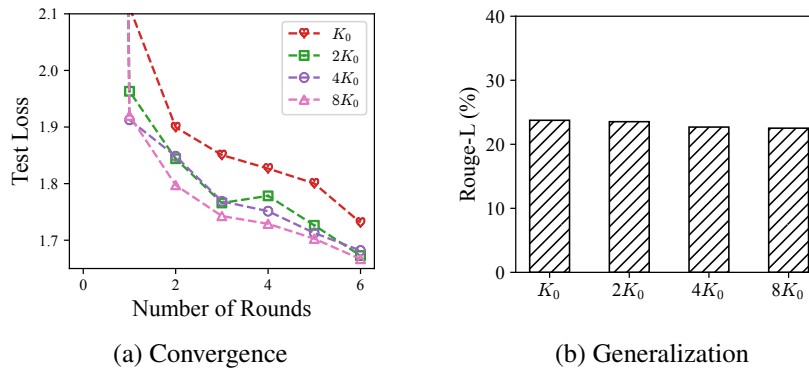

(a) Convergence

(b) Generalization

Figure 5: Convergence and generalization of Ferret under varying $K$ on Natural Instructions with DataJuicer-1.3B where $2K_0$ corresponds to the communication cost of $7.8 \times 10^3$ per round in Tab. 4.

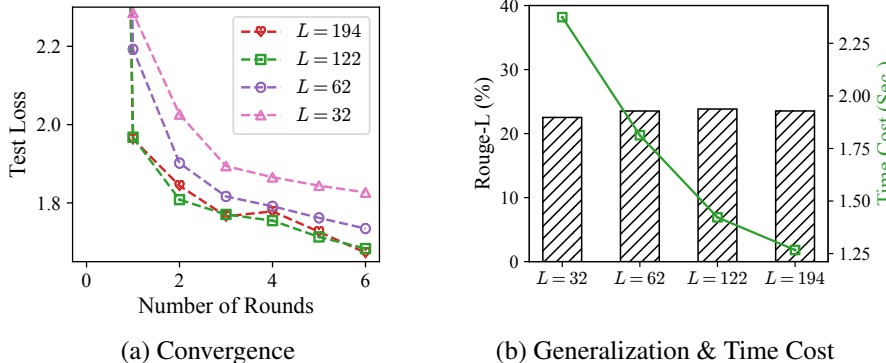

(a) Convergence

(b) Generalization & Time Cost

Figure 6: Convergence, generalization, and projection time cost per round of Ferret under varying $L$ on Natural Instructions with DataJuicer-1.3B where $L = 194$ is applied in our Tab. 2.

rate with respect to the dimension $d$ between our (8) method and the ZO method. Interestingly, Fig. 4 (b) shows that both methods achieve the same reconstruction accuracy rate with respect to $d$. More importantly, when $d$ becomes large, the accuracy rate is approximately linear, which aligns with the theoretical insights provided in Thm. 2.

**Reconstruction Accuracy of Ferret under Varying $L$.** In Fig. 4 (c), we present the reconstruction accuracy (measured by cosine similarity) and computational complexity (measured by time cost) for the $d = 5.12 \times 10^5$-dimensional gradient of function $F(\boldsymbol{x}) = \sum_{i=1}^{d} \sin^2(x_i)$ at a randomly sampled input $\boldsymbol{x}$, under varying $L$ of the same number of dimensions and $K = 512$, using our method in (8). The goal is to study the impact of block size $L$ on our (8). Notably, Fig. 4 (c) shows that our block-wise reconstruction (8) significantly reduces computational complexity (in line with Prop. 1), while maintaining consistent reconstruction accuracy as $L$ increases. These results further verify the efficacy of our block-wise reconstruction (8).

## C.5 ABLATION STUDIES ON CONVERGENCE AND GENERALIZATION

**Convergence and Generalization of Ferret under Varying $K$.** In Fig. 5, we present the convergence and generalization of Ferret under varying $K$ on the Natural Instructions dataset with DataJuicer-1.3B, using the same experimental setup as described in Appx. C.1. Notably, Fig. 5 shows that: *(a)* a larger number of random bases (i.e., a larger $K_0$) generally leads to improved convergence, while the generalization performance remains comparable; *(b)* $2K_0$ already provides compelling convergence and generalization performance, and further increasing $K$ yields only marginal improvements in convergence; and *(c)* a slight decrease in generalization performance as $K$ increases is likely due to the reduced regularization effect from noisy gradients.

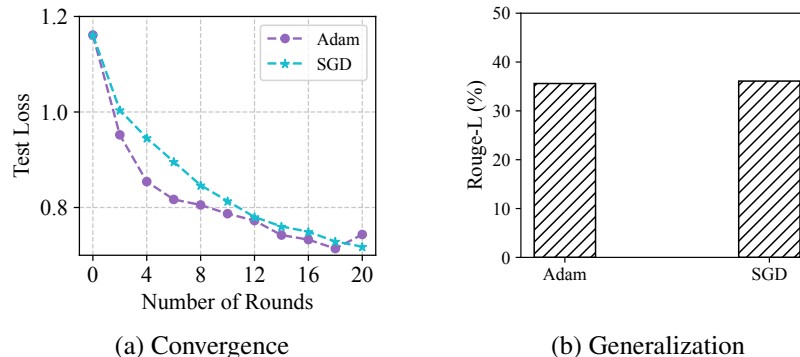

(a) Convergence  (b) Generalization

Figure 7: Convergence and generalization of Ferret under varying optimizers for local updates.

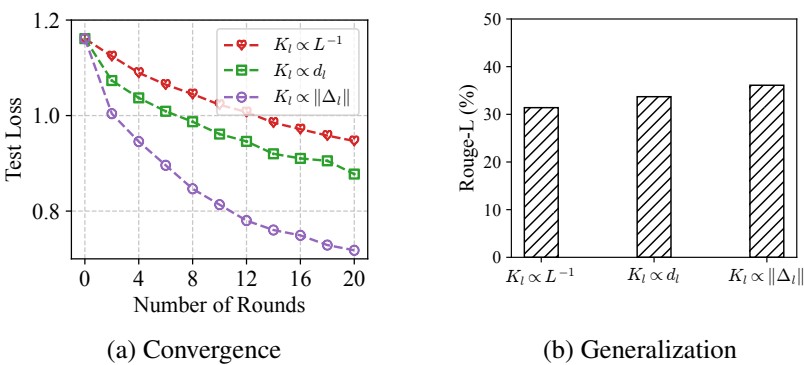

(a) Convergence  (b) Generalization

Figure 8: Convergence and generalization of Ferret under varying allocation scheme of $K$ for our block-wise reconstruction, in which $K_l \propto \|\Delta_l\|$ corresponds to our results in Sec. 5.

**Convergence and Generalization of Ferret under Varying $L$.** In Fig. 6, we present the convergence, generalization, and projection time cost of Ferret under varying block sizes $L$ on the Natural Instructions dataset with DataJuicer-1.3B, using the same experimental setup as described in Appx. 21. Notably, Fig. 6 shows that increasing the number of blocks (i.e., a larger $L$) leads to improved convergence and reduced time cost for projection and reconstruction, while the generalization performance remains comparable. This improved convergence is likely due to the logarithmic term in the reconstruction error of our (7), as a larger number of blocks reduces the dimensionality of each block, thereby minimizing reconstruction error. In addition, the reduced time cost aligns with our analysis in Sec. 4.1 and the empirical results shown in Fig. 4(c), further highlighting the efficacy of our block-wise reconstruction method (8).

**Convergence and Generalization of Ferret under Varying Optimizers.** In Fig. 7, we present the convergence and generalization of Ferret under different optimizers for its local updates, using the same experimental setup described in Appx. C.1. Notably, Fig. 7 demonstrates that Ferret achieves faster convergence with an improved optimizer (e.g., Adam vs. SGD) while maintaining comparable generalization performance. These findings further support the adaptability of Ferret, as discussed in Sec. 4.3.

**Convergence and Generalization of Ferret under Varying Allocation of $K$.** In Fig. 8, we present the convergence and generalization of Ferret under different allocation schemes for $K$ in our block-wise reconstruction, using the same experimental setup described in Appx. C.1, where $K_l \propto \|\Delta_l\|$ corresponds to our results in Sec. 5. Notably, Fig. 7 shows that Ferret achieves both faster convergence and improved generalization performance by following the best practices guided by Prop. 2. These findings therefore validate the significance and correctness of our Prop. 2.

