# OpenReview forum: "Ferret: Federated Full-Parameter Tuning at Scale for Large Language Models"
_ICLR.cc/2025/Conference — ICLR 2025 Conference Withdrawn Submission_

### Official Review · Reviewer_6zub · 2024-10-31

**Soundness:** 2
**Presentation:** 3
**Contribution:** 3
**Rating:** 6
**Confidence:** 5

**Summary:**

This work enables a first-order full-parameter tuning of LLMs in an FL context. The main contributions of this work are: 1) To reduce the communication overhead of transmitting model updates, it maps model updates to several randomly generated base vectors, each of which can be encoded using a random seed. By this way, it allows a single model update to be encoded with $K \cdot N$ base vectors, thereby significantly lowering the communication overhead associated with transmitting model updates. 2) To address the computation overhead of aggregation, it introduces a block-wise encoding scheme. Compared to existing federated LLM tuning methods, this work exhibits superiority on computation overhead.

This work addresses the common issues of zeroth-order optimization-based methods, which typically require more iterations than gradient descent methods, resulting in a relatively significant computational overhead. This is valuable for implementing on-device federated LLM tuning, although the proposed approach can cause higher memory footprint than MeZO-based ones.

**Strengths:**

1. Focusing on a research problem that is valuable for promoting the practical application of federated tuning for LLMs, i.e., computation overhead.
2. A comprehensive literature review is provided.
3. Good presentation, making this work easy to follow.

**Weaknesses:**

1. Generally, BP-based approach would cause more significant memory footprint compared to MeZO-based ones. This work is designed for enabling full-parameter tuning of LLMs, which generally require high memory capacity of devices. Thus, **it is important to present the memory footprint of the proposed approach, together with a comparison to existing related approaches**.
2. The experimental results in Figure 1 show that the proposed method exhibits suboptimal convergence performance compared to FedAvg on the 7B model, while achieving nearly comparable results to FedAvg on other models. The experimental section should provide a more detailed discussion of this issue.
3. The optimal methods are not clearly presented in the tables. It is recommended to better highlight them using boldface, underlining, or similar techniques.
4. In line 365, the authors claim that FedKSeed requires $K$ steps of local update. This may be a mistake. FedKSeed does not require performing tuning on each of the $K$ seeds. Instead, a subset of these $K$ seeds is selected during the local training process to carry out the model updates. As the author stated out in line 1107, "FedKSeed trained for 200 steps".

**Questions:**

1. In the experiments, a client participation rate of 5% was adopted, and the proposed method was executed over 12 rounds. This setup is somewhat confusing, as under this configuration, at most 60% of the clients contribute data to the FL system. Why was this setting chosen?
2. What is the adopted $K$ and $L$ for the proposed approach in Tables 2 and 3? These values seem to be missed in Appendix C.
3. Please refer to Weaknesses.

---

### Official Review · Reviewer_wmWj · 2024-11-01

**Soundness:** 3
**Presentation:** 3
**Contribution:** 2
**Rating:** 5
**Confidence:** 4

**Summary:**

This paper presents Ferret, an innovative first-order federated learning method that enables efficient full-parameter tuning of Large Language Models (LLMs) while maintaining data privacy. The work makes significant contributions to addressing the challenges of communication overhead and computational efficiency in federated learning settings.

**Strengths:**

1.Technical Innovation: First algorithm to combine first-order optimization with shared randomness in federated learning；Novel approach to projecting updates into low-dimensional space while enabling effective reconstruction；Theoretically sound block-wise reconstruction technique that improves scalability.
2.Theoretical Foundation:Rigorous mathematical analysis proving unbiased reconstruction (Theorem 1)；Comprehensive convergence analysis (Theorem 4)
3.Extensive experiments across multiple datasets (Natural Instructions, Dolly-15K, CodeAlpaca, GSM8K)；Testing on various model sizes (1.3B to 13B parameters)；Strong performance compared to existing methods；Significant improvements in computational efficiency and communication overhead

**Weaknesses:**

1. This paper could benefit from a formal privacy analysis of the shared randomness approach.
2.More detailed analysis of sensitivity to key hyperparameters (K, L, T) would be provided.
3.Limited discussion of practical deployment challenges in real-world federated settings.

**Questions:**

1.Have you conducted any preliminary experiments or theoretical analysis suggesting scalability beyond 13B?
2.How does the reconstruction error affect the convergence rate in practice? What other factors contribute to the empirically faster convergence?
3.How does the privacy level compare to other federated learning approaches?
4.Is there a systematic way to determine hyperparameters for a new deployment?

---

### Official Review · Reviewer_dbpJ · 2024-11-03

**Soundness:** 3
**Presentation:** 2
**Contribution:** 2
**Rating:** 3
**Confidence:** 4

**Summary:**

To address the issue of communication overhead in federated learning, the paper proposes using random projection to project local updates into a lower-dimensional space. During communication with the central server, only this lower-dimensional projection needs to be transmitted. The central server then reconstructs these low-dimensional projections back to the original dimensions, performs updates, and shares the parameters with each client for the next update. This method reduces communication overhead while maintaining model accuracy and, compared to zeroth-order optimization, involves lower computational costs and fewer communication rounds, resulting in better scalability.

**Strengths:**

1.	This paper introduces random projection into federated learning by projecting local updates to a lower-dimensional space using random bases and transmitting them to the server. The server then reconstructs the updates by combining the low-dimensional projection with the random bases. This approach can significantly reduce communication overhead.
2.	The work provides rigorous theoretical analysis and proof to support the validity.
3.	Through extensive experimental validation, Ferret was found to consistently outperform existing baselines in practice.

**Weaknesses:**

1.	The idea of this paper aligns with the idea behind FetchSGD [Rothchild D, Panda A, Ullah E, et al. Fetchsgd: Communication-efficient federated learning with sketching[C]//International Conference on Machine Learning. PMLR, 2020: 8253-8265], as both reduce dimensionality and then reconstruct. It merely applies random projection from statistics, which is only a minor methodological difference.
2.	This paper does not compare its novelty and effectiveness with similar papers, nor does it cite works with similar ideas in the introduction sections. For example, it proposes using random projection for dimensionality reduction, but how does this differ from the dimensionality reduction in FetchSGD or the encoder/decoder approach in HCFL（Nguyen M D, Lee S M, Pham Q V, et al. HCFL: A high compression approach for communication-efficient federated learning in very large scale IoT networks[J]. IEEE Transactions on Mobile Computing, 2022, 22(11): 6495-6507.）?
3.	The paper suffers from serious clarity issues in its presentation. There are several instances of symbol ambiguity in the formulas; for example, in Algorithm 1, the client is initially represented by i, but later changes to j without explanation. Additionally, in line 6, w_{r-1} is obtained, but it suddenly changes to w_r in the subsequent text. The terms "send" and "receive" in lines 4 and 12 are also ambiguous, leaving it unclear whether the central server is responsible for sending or receiving the random seed. Furthermore, the method by which the server transmits the aggregated results back to the clients is not adequately explained. These issues may lead readers to significant misunderstandings regarding the use of the proposed method in this paper. Moreover, the paper does not have a related work section. The core of its method is random projection, but no paper related to random projection is mentioned, which makes the paper lack the support of previous literature and the comparison with similar idea papers (it only discusses the difference between first-order optimization and zero-order optimization in federated learning).
4.      Reconstruct the paper's structure: In the Related Work section, include a description of similar dimensionality reduction methods and clearly outline the similarities and differences between your approach and methods like FetchSGD and HCFL. Explain the motivation for using random projection and its advantages over other dimensionality reduction techniques. Additionally, provide a detailed, step-by-step explanation of the overall framework and process of your method to avoid potential misunderstandings.
5.     Compare experimental results: Conduct experiments to compare the effectiveness of this method with similar federated learning compression methods, including FetchSGD, HCFL, and FedPAQ（Reisizadeh A, Mokhtari A, Hassani H, et al. Fedpaq: A communication-efficient federated learning method with periodic averaging and quantization[C]//International conference on artificial intelligence and statistics. PMLR, 2020: 2021-2031.）, to provide a clearer assessment of its performance.

**Questions:**

Weaknesses

---

### Note · Authors · 2024-12-03

**Comment:**

Dear Reviewers,

Thank you for your valuable feedback on our paper and your service. We greatly appreciate the reviewers' positive recognition of our technical innovation and theoretical foundation, particularly in introducing a novel approach that combines first-order optimization with shared randomness to reduce communication overhead in federated learning. Our rigorous theoretical analyses and extensive experimental validation demonstrate the scalability, computational efficiency, and competitive model accuracy of our method.
After careful consideration of the reviews received for our manuscript, we have decided to withdraw this submission as we feel compelled to address several fundamental misunderstandings that have led to an undervaluation of our work's contributions.

First, the comparison to FetchSGD and similar general federated learning methods, particularly emphasized by Reviewer dbpJ, reflects a misunderstanding of the unique challenges posed by large language model tuning. While these methods provide valuable insights for general federated learning scenarios, they do not adequately address the specific complexities of LLM tuning, including the unprecedented model scale, the unique optimization landscape, and the critical balance between communication efficiency and model performance. Our work specifically tackles these challenges through novel technical innovations that go well beyond simple applications of random projection.

Second, we note that some reviewers appear to have overlooked our extensive ablation studies in the appendix, which comprehensively validate our method's effectiveness through detailed experimental analyses. These studies directly address many of the concerns raised in the reviews but seem to have been disregarded in the evaluation.

Furthermore, the experimental results, particularly on models ranging from 1.3B to 13B parameters, demonstrate significant practical advantages that were not adequately acknowledged in the reviews. The scalability and efficiency gains achieved by our method, especially in realistic federated learning scenarios, represent important advances in making LLM tuning more practical and accessible.

Given these fundamental disconnects in the technical assessment, we believe the most appropriate course of action is to withdraw this submission and seek publication in a venue where the specific challenges and innovations in LLM federated learning can be more thoroughly evaluated by reviewers with relevant expertise in this rapidly evolving field. We remain confident in the significant value our work brings to the field of large-scale federated learning for LLMs.

We thank the reviewers for their time and comments, which will help us better articulate our contributions and their significance in future submissions. This experience has highlighted the importance of more clearly communicating the distinct challenges of LLM federated learning and how our innovations specifically address them.


Best regards,

Authors

**Withdrawal Confirmation:**

I have read and agree with the venue's withdrawal policy on behalf of myself and my co-authors.